# Distributed feature representations of natural stimuli across parallel retinal pathways

Jen-Chun Hsiang [1,4], Ning Shen[1,4], Florentina Soto [1] &
Daniel Kerschensteiner [1,2,3] ✉

How sensory systems extract salient features from natural environments and organize them across neural pathways is unclear. Combining single-cell and population two-photon calcium imaging in mice, we discover that retinal ON bipolar cells (second-order neurons of the visual system) are divided into two blocks of four types. The two blocks distribute temporal and spatial information encoding, respectively. ON bipolar cell axons co-stratify within each block, but separate laminarly between them (upper block: diverse temporal, uniform spatial tuning; lower block: diverse spatial, uniform temporal tuning). ON bipolar cells extract temporal and spatial features similarly from artificial and naturalistic stimuli. In addition, they differ in sensitivity to coherent motion in naturalistic movies. Motion information is distributed across ON bipolar cells in the upper and the lower blocks, multiplexed with temporal and spatial contrast, independent features of natural scenes. Comparing the responses of different boutons within the same arbor, we find that axons of all ON bipolar cell types function as computational units. Thus, our results provide insights into the visual feature extraction from naturalistic stimuli and reveal how structural and functional organization cooperate to generate parallel ON pathways for temporal and spatial information in the mammalian retina.

Sensory systems extract salient features of the world and process them in parallel pathways. Significant progress has been made in understanding the feature preferences of individual pathways for simple artificial stimuli[1–5]. However, how features are extracted from complex natural stimuli and how feature representations are organized across parallel pathways is unclear.

In the visual system, parallel processing begins at the first synapse, where photoreceptor signals are distributed to multiple bipolar cells[6,7]. Mice have 15 bipolar cell types[8–12]; most are conserved in primates, including humans[13–16]. One of the bipolar cells exclusively contacts rods; the other 14 receive input from cones[17,18]. Cone bipolar cells are grouped into eight ON and six OFF types (referred to as ON and OFF bipolar cells hereafter), which depolarize to light increments and decrements, respectively[8–12]. The split into ON and OFF channels helps efficiently encode luminance distributions in natural scenes[19–22]. The functional divergence and organization within both channels are not well understood.

For artificial stimuli, some studies noted varying preferences for spatial and temporal contrast among ON and OFF bipolar cells[11,23,24], while others reported complex feature preferences of specific bipolar cell types (e.g., direction and orientation selectivity)[25,26]. What features mammalian bipolar cells extract from naturalistic stimuli and how

[1]Department of Ophthalmology and Visual Sciences, Washington University School of Medicine, St. Louis, MO 63110, USA. [2]Department of Neuroscience, Washington University School of Medicine, St. Louis, MO 63110, USA. [3]Department of Biomedical Engineering, Washington University School of Medicine, St. Louis, MO 63110, USA. [4]These authors contributed equally: Jen-Chun Hsiang, Ning Shen. ✉e-mail: kerschensteinerd@wustl.edu

features of natural environments are organized across parallel bipolar pathways remains unknown.

Laminar feature maps organize information throughout visual systems across evolution[4,27–29]. ON and OFF bipolar cell axons occupy the outer 2/5 and inner 3/5, respectively, of the mouse retina's inner plexiform layer (IPL)[6]. Rod bipolar cells target the innermost stratum of the IPL, conveying dim-light information[6,15]. In addition, two color-selective channels (one OFF and one ON) are found at the edges of the IPL[6,30,31]. If and how other visual features encoded by bipolar cells (e.g., temporal contrast, spatial contrast, and motion) are mapped across the depth of the IPL is unclear.

Bipolar cell axons integrate photoreceptor signals relayed by their dendrites from the outer retina with local inputs from amacrine cells, a diverse class of interneurons in the inner retina[6,11,32]. It has been suggested that amacrine cell influences diversify the responses within a single bipolar cell axon[25,33,34]. Subcellular processing could expand the number of bipolar cell pathways in the inner retina. Whether individual bipolar cell axons encode different information in different presynaptic boutons or function as computational units under naturalistic stimulus conditions remains to be tested.

Here, we combine viral and transgenic labeling with two-photon imaging to understand how the complement of ON bipolar cell types encodes naturalistic stimuli and how their axons organize features extracted from natural environments in the IPL, from single arbors to neural populations.

## Results

### Labeling and classification of ON bipolar cell types

To compare visual processing across ON bipolar pathways and delineate computational units in the inner retina, we recorded axonal calcium transients in morphologically identified bipolar cell types with subcellular resolution by two-photon imaging. We injected adeno-associated viruses (AAVs) expressing Cre recombinase from *Grm6* promoter elements (*AAV-Grm6-Cre*) intravitreally into GCaMP6f (i.e., a genetically encoded calcium indicator) reporter mice (*Ai148*)[35,36]. This strategy uncoupled labeling density (controlled by viral infection) and specificity (controlled by the viral promoter) from expression levels (controlled by the reporter mice) and enabled sparse but intense GCaMP6f expression in ON bipolar cells (Fig. 1a, c).

We restricted recordings to isolated bipolar cells that could be morphologically classified, limited the imaging time per retina to avoid tissue damage from laser scanning[37], and set stringent thresholds for repeat reliability of light responses. Thus, we analyzed the responses of 57 ON bipolar cells, representing all types, in 41 retinas of 378 AAV-injected mice (Fig. 1b).

To compare visual processing across bipolar cell types without circularity, we classified them independently of their function. After two-photon calcium imaging, we fixed retinas, stained them for cone arrestin (CAR) and choline acetyltransferase (ChAT), and analyzed the morphology and connectivity of bipolar cell dendrites and axons (Fig. 1c–f). Referencing large-scale anatomical surveys[8,10,12,18], we distinguished BC5o, BC5i, BC5t, XBC, BC6, BC7, and BC8/9 bipolar cells by their dendrite and axon territories (Fig. 1d, e), the numbers of cones contacted (Fig. 1d), and axonal stratification patterns (Fig. 1f) with a classification tree (Supplementary Fig. 1). We combined results from BC8 and BC9 because we could not differentiate them anatomically, and their responses in the ventral retina, the location of our recordings, are equally dominated by short-wavelength-sensitive S-opsin[30,31,38,39].

Together with serial electron microscopy (EM) reconstructions[8,12,40], our data suggest that ON bipolar cell axons form two laminar blocks, one above the ON ChAT band (i.e., the upper block) encompassing BC5o, BC5i, BC5t, and XBC, and one below the ON ChAT band (i.e., the lower block) encompassing BC6, BC7, and BC8/9. The stratification patterns of ON bipolar cell axons overlapped extensively within each block but not across (Fig. 1g).

### Temporal frequency tuning across ON bipolar pathways

ON bipolar cells vary in their preferences for temporal stimulus contrast[11,23]. We analyzed the responses of the full complement of ON bipolar cell axons to sinusoidal luminance fluctuations of varying temporal frequencies (Fig. 2). The frequency-response curves were normalized by the profile of the GCaMP6f indicator (Supplementary Fig. 2). We used simulations to confirm that even though the indicator attenuates amplitudes at higher frequencies, we could accurately assess responses to 8 Hz stimuli and slower fluctuations (Supplementary Fig. 3). We first presented fluctuations in a 150 μm spot restricted to the receptive field center (i.e., the region in which excitatory outweigh inhibitory influences)[41]. The frequency-response functions of most ON bipolar cell types did not differ significantly, with two exceptions: XBC and BC5t (Fig. 2a, b). XBC preferred higher frequency stimuli than the rest of the ON bipolar cells (Fig. 2a, b, one-sided Wilcoxon rank sum test, $p < 0.001$, $r = 0.65$, $z = 4.08$, Supplementary Fig. 3), whereas BC5t preferred lower frequency stimuli than the rest (Fig. 2a, b, one-sided Wilcoxon rank sum test, $p < 0.01$, $r = -0.42$, $z = -2.59$, Supplementary Fig. 3). Some ON bipolar cell types (incl. BC5t) exhibited frequency-dependent suppressive baseline (i.e., F0) responses, whose mechanisms remain to be identified.

The temporal tuning of ON bipolar cell axons can be shaped by their morphology. In zebrafish and goldfish bipolar cells, smaller presynaptic terminals (i.e., boutons) prefer faster stimulus frequencies than larger boutons due to changes in the surface/volume ratio[42]. Yet the ROIs of XBC, which prefer the fastest frequency among ON bipolar cells, did not differ significantly in size from ROIs of other ON bipolar cell types, and the ROIs of BC5t, which prefer the slowest stimulus frequencies, were smaller than the ROIs of other ON bipolar cells (Supplementary Fig. 4). Moreover, when we divided ROIs into small and large (50/50) groups across (Supplementary Fig. 4) or within types (Supplementary Fig. 5), we observed no differences in their temporal tuning. Thus, the cell-type-specific differences in temporal tuning of mouse ON bipolar cells are not accounted for by their morphology.

To examine the influence of the receptive field surround (i.e., the region in which inhibitory outweigh excitatory influences)[41] on the temporal tuning of ON bipolar cells, we presented sinusoidal fluctuations in an 800 μm spot encompassing center and surround. At low temporal frequencies (≤2 Hz), 800 μm stimuli elicited ON bipolar cell responses with similar amplitudes as the 150 μm stimuli (Fig. 2c, Supplementary Fig. 6). However, the responses to these stimuli were phase-shifted (Supplementary Fig. 6) reflecting the different luminance preferences of the receptive field center (ON) and surround (OFF). Responses to large stimuli also extended to higher temporal frequencies (Fig. 2c, comparing 0.5–2 Hz and 4–8 Hz, one-sided Wilcoxon signed-rank test, $p < 0.01$, $r = -0.45$, $z = -2.9$)[11]. This high-frequency enhancement was most pronounced for BC5t, BC7, and BC8/9.

Thus, the temporal frequency-response functions of six of the eight parallel ON bipolar pathways overlap with two opposing outliers (BC5t and XBC). Both outliers are localized in the upper block, generating diversity in temporal processing within this block, whereas temporal processing is uniform in the lower block. Interactions between receptive field centers and surrounds extend response functions to higher frequencies in a cell-type-specific manner.

### Divergence of temporal and spatial information in ON bipolar pathways

To gain further insights into the ON bipolar cells' processing of temporal and spatial information, we analyzed responses to light steps presented in spots of varying size (Fig. 3a, b). In mammalian retinas, bipolar cell axons with transient responses are thought to stratify

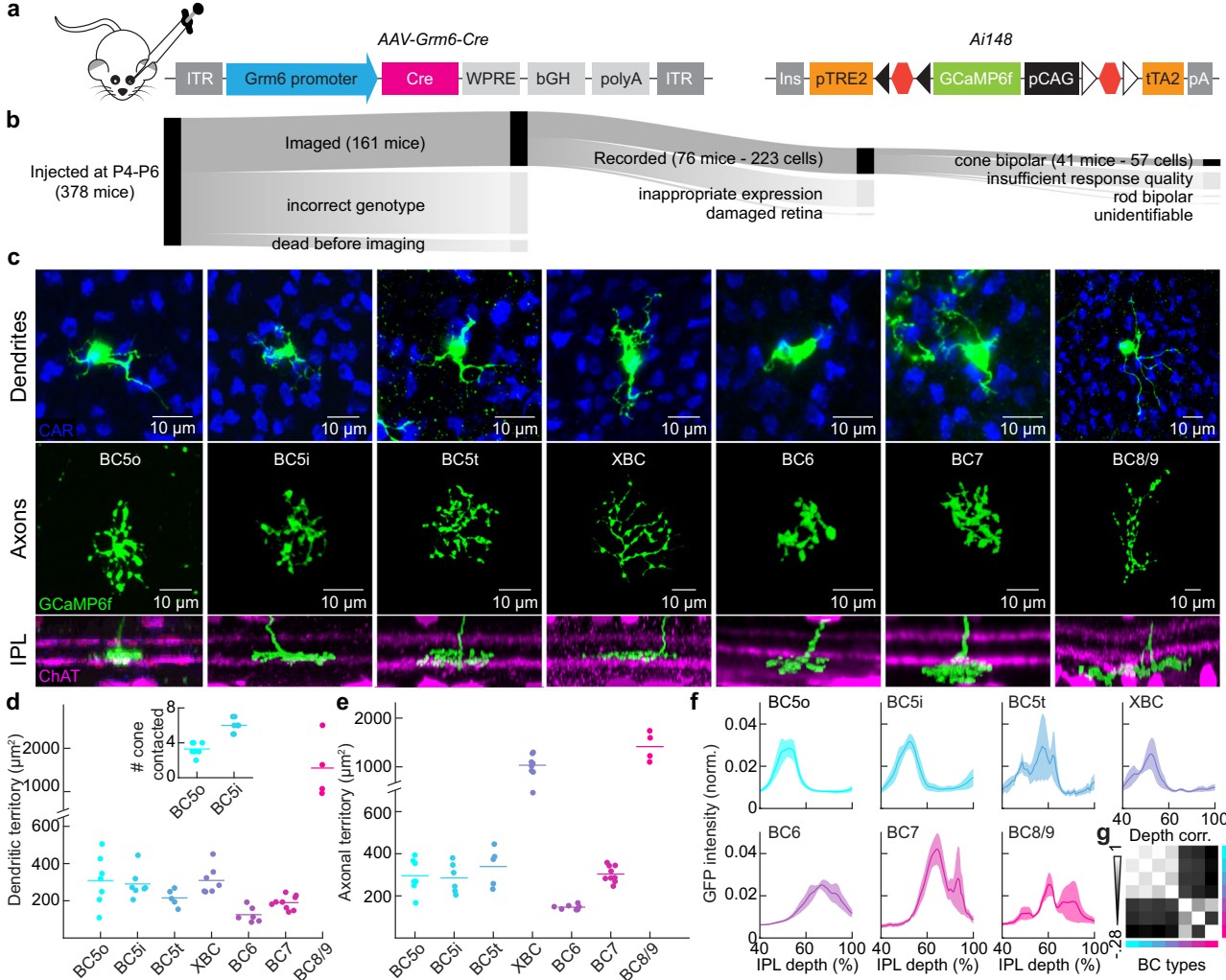

**Fig. 1 | Labeling and morphological classification of ON bipolar cells. a** Diagram illustrating intravitreal injection (left), the adeno-associated virus (AAV) constructs carrying *Grm6* promoter elements, Cre recombinase genes (middle), and the conditional allele of the *Ai148* mouse strain (right). This allele features a *TRE2* promoter, *loxP*-flanked *STOP* cassettes followed by *GCaMP6f*, and then a *CAG* promoter, *lox2272*-flanked *STOP* cassette, and *tTA2*. **b** The main dark-shaded branch depicts the number of mice and bipolar cells that met quality-control criteria during each data collection phase, while the lighter subsidiary branches detail the criteria for data rejection, with branch widths representing the number of rejections; the widths of the two blocks at left and center are proportional to the numbers of mice, right block is proportional to the number of cells. **c** Top, middle, and bottom images showcase *en-face* views of dendrites and axons, and side view of the inner plexiform layer (IPL), respectively. Co-stains include cone arrestin (CAR) in blue,

labeling cone photoreceptor, and choline acetyltransferase (ChAT) in magenta, marking starburst amacrine cells. Scale bars (shared for axon and IPL images) are included. The swarm chart shows dendritic territories with (**d**) listing bipolar cell types and counts as BC5o ($n = 7$), BC5i ($n = 7$), BC5t ($n = 5$), XBC ($n = 7$), BC6 ($n = 6$), BC7 ($n = 9$), BC8/9 ($n = 4$) and axonal territories (**e**) as BC5o ($n = 7$), BC5i ($n = 6$), BC5t ($n = 5$), XBC ($n = 8$), BC6 ($n = 7$), BC7 ($n = 9$), BC8/9 ($n = 4$). BC5o and BC5i differentiation is based on the number of cones contacted[18], shown in the inset. **f** IPL stratification profiles of all ON bipolar cell types, BC5o ($n = 6$), BC5i ($n = 5$), BC5t ($n = 5$), XBC ($n = 6$), BC6 ($n = 6$), BC7 ($n = 8$), BC8/9 ($n = 3$). The graph has seven color-coded traces, indicating the mean (±SEM). **g** This denotes the correlation coefficient of paired bipolar cell stratification profiles from connectomic reconstructions[8]. Source data for this figure are provided as a Source Data file.

toward the center of the IPL, and bipolar cell axons with sustained responses toward the IPL borders[6,11,43–49]. Our data supports this overall trend, but rather than a gradual progression from transient to sustained, it revealed discontinuous patterns. First, the responses of axons in the upper block were, on average, more transient than those in the lower block (Fig. 3a–c, one-sided Wilcoxon rank sum, upper block: $n = 27$, lower block: $n = 20$, $p < 0.001$, $r = 0.61$, $z = 4.19$). Second, response kinetics were uniform in the lower block (Kruskal-Wallis one-way ANOVA, $p = 0.46$, $\chi^2 = 1.57$, $\eta^2 = 0.079$, df = 2) but varied between bipolar cell types in the upper block (Kruskal-Wallis one-way ANOVA, $p < 0.001$, $\chi^2 = 21.6$, $\eta^2 = 0.8$, df = 3). BC5t axons in the upper block exhibited slow and sustained responses (Fig. 3a–c), matching their lower frequency cut-offs for sinusoidal fluctuations (Fig. 2a, b). BC5t axons co-stratify with XBC axons, which had the most transient

responses of all ON bipolar cells (Fig. 3a–c), in line with their filtering of sinusoidal fluctuations (Fig. 2a, b).

The opposite picture emerged for spatial response profiles (Fig. 3a–c). Axons of all bipolar cell types in the upper block (BC5o, BC5i, BC5t, and XBC) showed similarly strong surround suppression, with responses to 800 µm spots reduced to 34% ± 9.8% of responses to 150 µm spots (Fig. 3a). By contrast, spatial receptive fields diverged among axons in the lower block (Kruskal-Wallis one-way ANOVA, $p < 0.01$, $\chi^2 = 9.64$, $\eta^2 = 0.48$, df = 2). BC7 axons exhibited extreme surround suppression, inverting their responses to spots ≥400 µm in diameter (Fig. 3a–c), whereas BC6 and BC8/9 showed more moderate surround suppression than BC7 and axons in the upper block (Fig. 3a–c). The diversity of temporal and spatial tuning in the upper and lower block, respectively, was further supported by pairwise

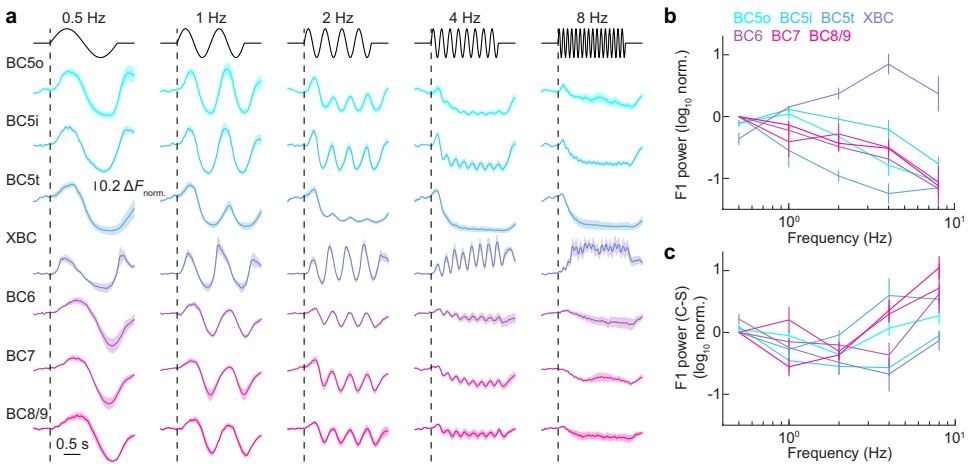

**Fig. 2 | Frequency responses of ON bipolar cells. a** ON bipolar cell types were exposed to stimuli featuring 2-second sinusoidal contrast fluctuations, with cell types and counts as follows: BC5o ($n = 7$), BC5i ($n = 7$), BC5t ($n = 4$), XBC ($n = 7$), BC6 ($n = 6$), BC7 ($n = 6$), and BC8/9 ($n = 5$). Shaded regions denote the mean ± SEM for each type, differentiated by unique colors. **b** Illustration of aggregate data of F1 power from 0.5 Hz to 8 Hz for each cell type, adjusted according to the GCaMP6 response function (see Methods); lines indicate the mean, and error bars signify ±SEM (same for (**c**)). Sample sizes correspond to those in (**a**). **c** Plots of the difference in the F1 power of responses to 150 μm (C: center, in diameter) and 800 μm (S: surround) spots across bipolar cell types, with frequencies from 0.5 to 8 Hz. Sample sizes correspond to those in (**a**). Source data for this figure are provided as a Source Data file.

comparisons of differences in response transience and surround strength between different-type bipolar cells in each block (Fig. 3d).

Some ON bipolar cells express voltage-gated sodium channels[9,50,51], raising the question of whether fire action potentials (i.e., spikes)[43]. If ON bipolar cells spike, we would expect response amplitudes to have binomial distributions and spontaneous spike events to occur occasionally when the light is off[43]. All ON bipolar cell types showed continuous response amplitude distributions to spots of varying size, and we observed no spike events during the OFF phase of these stimuli across 3215 trials in 49 bipolar cells (Supplementary Fig. 7). Thus, the ON bipolar cells in our recordings did not spike.

When we measured the functional distances between ON bipolar cells in the spatiotemporal response maps (Fig. 3b), we found that most types differed significantly from each other (Fig. 3e). Similarly, most bipolar cells were correctly assigned to their morphological type by a functional distance-based classifier in a leave-one-out cross-validation test (Fig. 3f). The only outlier was BC5t, frequently misassigned to sustained types in the lower block. This could be corrected by including information about stratification depth in the classifier (Fig. 3f, g). The resulting simple classifier performed remarkably well (29–89% correct, chance level: 14.3%).

### The encoding space of ON bipolar cells for artificial stimuli

The ability to reliably identify ON bipolar cell types with a simple classifier derived from observations of isolated axons allowed us to record simultaneously from populations of bipolar cells labeled by crossing *Grm6-Cre* to *Ai148* mice[35,52] and address questions about their encoding space that require more data.

We segmented two-photon imaging series from the IPL of *Grm6-Cre Ai148* mice by finding peaks in the temporal standard deviation. We grew regions of interest (ROIs) radially from these peaks until they met pixels of neighboring ROIs or fell below a threshold for inclusion (Fig. 4a, median ROI size: 1.72 μm²). We measured the correlation of ROIs' responses to repetitions of the same stimulus and rejected non-responsive ROIs (i.e., ROIs with low repeat reliability). We then used the classifier derived from our recordings of morphologically identified cells to assign responsive ROIs to specific cell types (Fig. 4a). Statistical outliers with low classification probabilities were removed (<2% of ROIs). For each imaging series, ROIs assigned to the same cell type (by unsupervised k-means clustering) were combined and added as a single data point to our set (see Methods).

To analyze the encoding space across all ON bipolar cells in the augmented data set, combining single-cell and population recordings, we performed a nonmetric multidimensional scaling (MDS) analysis on the responses to spots of varying size. The first two principal coordinates accounted for most of the variation in the data (Supplementary Fig. 8). To test the potential impact of interexperimental variations (i.e., batch effects)[53] on this analysis, we split data sets into two and calculated the Jensen-Shannon divergence (JSD) of the resulting distributions (see Methods). Repeating this for 2000 different splits, we found that the JSDs did not differ significantly from zero ($p = 0.32$), indicating that batch effects did not shape the encoding space defined by the first two principal coordinates (Supplementary Fig. 9).

We next examined the positions of different cell types in the encoding space (Fig. 4b, c). Most cell types occupied distinct but overlapping regions, indicating that parallel pathways encode different information (Fig. 4b, c). To gain further insights into the features encoded in the space, we calculated indices of surround strength and response transience for each cell in the data set. Mapping surround strength and response transience onto the encoding space (Fig. 4d, e) revealed that they define nearly orthogonal axes organizing the data (Supplementary Fig. 10). To quantify how closely cell positions in the encoding space aligned with gradients in surround strength and response transience, we calculated the angles between the average vector of all pairs of cells ($n = 41,965$) vs. 90-pair subsamples (0.21%) in Monte-Carlo simulations (Fig. 4d, e). Most subsample vectors fell within 20° of the population (99.8% for surround strength and 76.4% for transience), indicating that the two axes explain most of the variation between ON bipolar cells and define their encoding space. Furthermore, the positions of bipolar cell types in the upper block (BC5o, BC5i, BC5t, and XBC) varied predominantly along the axis defined by response transience, whereas bipolar cell types in the lower block (BC6, BC7, and BC8/9) differed in their placement along the axis defined by surround strength (Fig. 4b–e).

Thus, an unbiased analysis of the encoding space of ON bipolar cells revealed that differences in their responses to an artificial stimulus are dominated by differences in spatial and temporal tuning and confirmed the organization of ON bipolar cells into two blocks that distribute encoding of temporal (the upper block) and spatial information (the lower block), respectively.

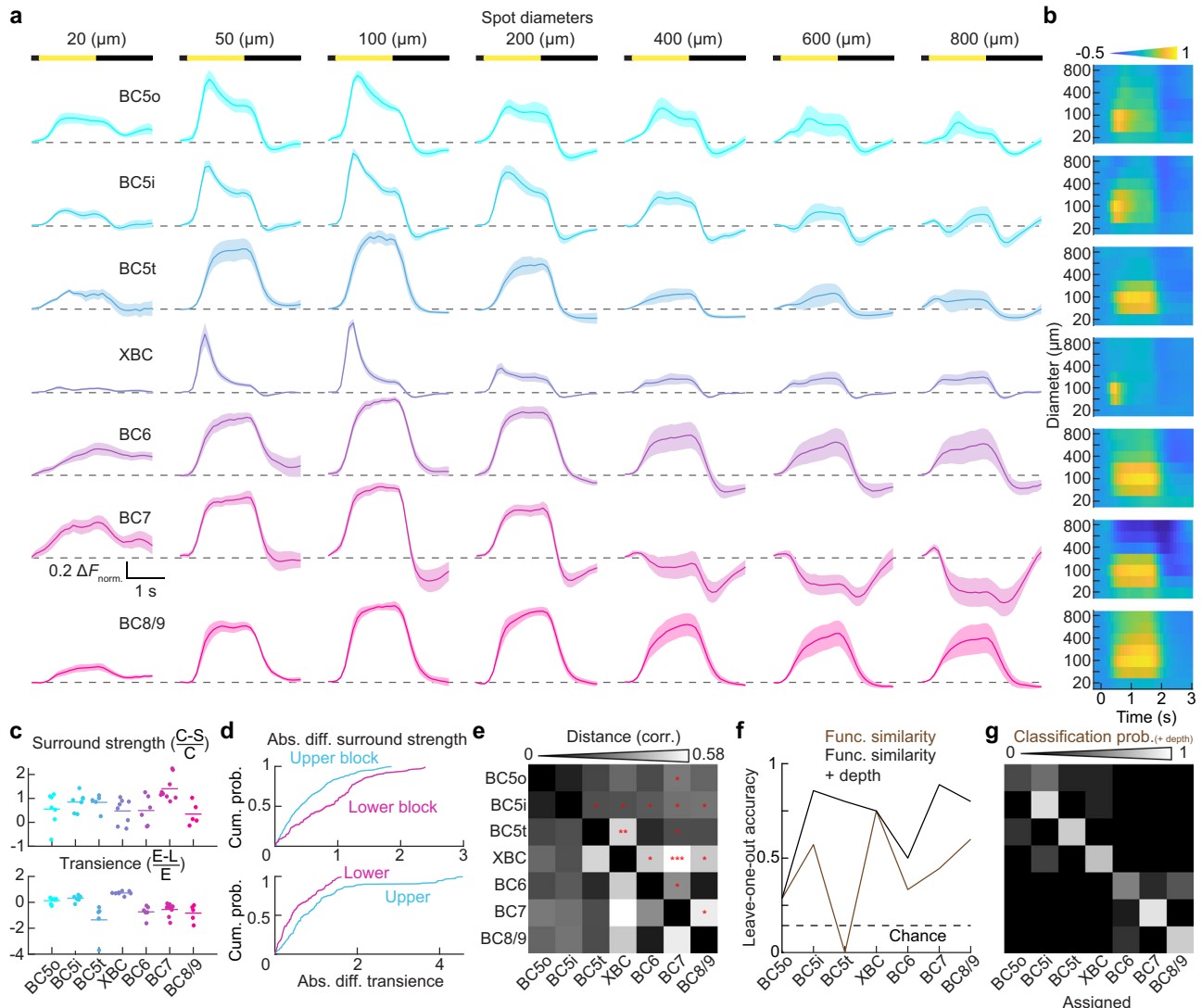

**Fig. 3 | Responses of ON bipolar cells to light steps in varying size spots.**
**a** Individual response traces for different bipolar cell types to square-wave modulated spots ranging from 20 to 800 μm in size. Each spot is presented at full brightness for 1.5 s following the background intensity, then at full darkness for an additional 1.5 s. The dashed line indicates a baseline, calculated as the average for 200 ms before the stimulus onset. Mean responses are represented by shaded areas, with the variation indicated by ±SEM. Different cell types are color-coded: BC5o ($n = 7$), BC5i ($n = 7$), BC5t ($n = 5$), XBC ($n = 8$), BC6 ($n = 6$), BC7 ($n = 9$), BC8/9 ($n = 5$). Each 'n' represents an individual cell from a separate animal. **b** Summary response plot derived from (**a**). Columns represent distinct time points, while rows correspond to individual tested spot sizes. The gradient colors, as specified at the top, denote the response amplitude. **c** Top: Summary data illustrating that surround strength primarily varies in the lower block (Kruskal-Wallis one-way ANOVA, $p < 0.0081$, $\chi^2 = 9.64$, $\eta^2 = 0.48$, df = 2) as opposed to the upper block (Kruskal-Wallis one-way ANOVA, $p = 0.49$, $\chi^2 = 2.4$, $\eta^2 = 0.089$, df = 2). The calculation for surround strength involves comparing responses to spots of receptive field center (C) and the surround (S). Bottom: Transience variation is evident in the upper block (Kruskal-Wallis one-wave ANOVA, $p < 0.001$, $\chi^2 = 21.6$, $\eta^2 = 0.8$, df = 3) but not significantly in the lower block (Kruskal-Wallis one-wave ANOVA, $p = 0.46$, $\chi^2 = 1.5$,

$\eta^2 = 0.079$, df = 2). Transience is calculated using early (E: 0.2–0.7 s after light onset) and late (L: 1.2–1.7 s after light onset) light responses. Sample size are identical to (**a**). **d** Cumulative distributions of differences in surround strength (top, Kolmogorov-Smirnov test, $p < 0.001$, $d = 0.32$) and response transience (bottom, Kolmogorov-Smirnov test, $p < 0.001$, $d = 0.28$) between different-type ON bipolar cell pairs in the upper block (cyan) and lower blocks (magenta). **e** Distance metric (expressed as $1 - correlation\ coefficient$) for each pair of bipolar cell types. Statistical significance in the upper triangle is denoted by asterisks (* $p < 0.05$, ** $p < 0.01$, *** $p < 0.001$). Two-sided permutation tests were performed, and their p-values were adjusted for multiple testing by controlling the false discovery rate (FDR) at a threshold of <0.05. **f** Leave-one-out classification tests based on functional similarity are highlighted in light brown font, whereas those incorporating depth information are in black. **g** Details the corresponding classification errors. With the maximum probability approach, each test cell is assigned one of the seven possible types, yielding an expected chance level of 14.3%. **g** Classification probabilities for each bipolar cell type. Rows represent the test cell under consideration, while columns indicate classification outcomes. Diagonal entries reflect correct classifications, with other areas indicating misclassifications. Source data for this figure are provided as a Source Data file.

## The cell-type-specific responses of ON bipolar cells to naturalistic stimuli

To explore how ON bipolar cells encode naturalistic stimuli individually and as a population, we first acquired movies from a mouse's perspective. We mounted a camera (1080 p, 60 fps) on a rolling support frame an inch off the ground pitched at an angle matching the mouse's fixation position (Supplementary Fig. 11)[54]. We then moved the camera at speeds matching the locomotion of mice through diverse environments, interspersed by periods of quiescence (Supplementary Fig. 11)[55,56]. We calibrated and corrected distortions in visual angle from the camera (Supplementary Fig. 11).

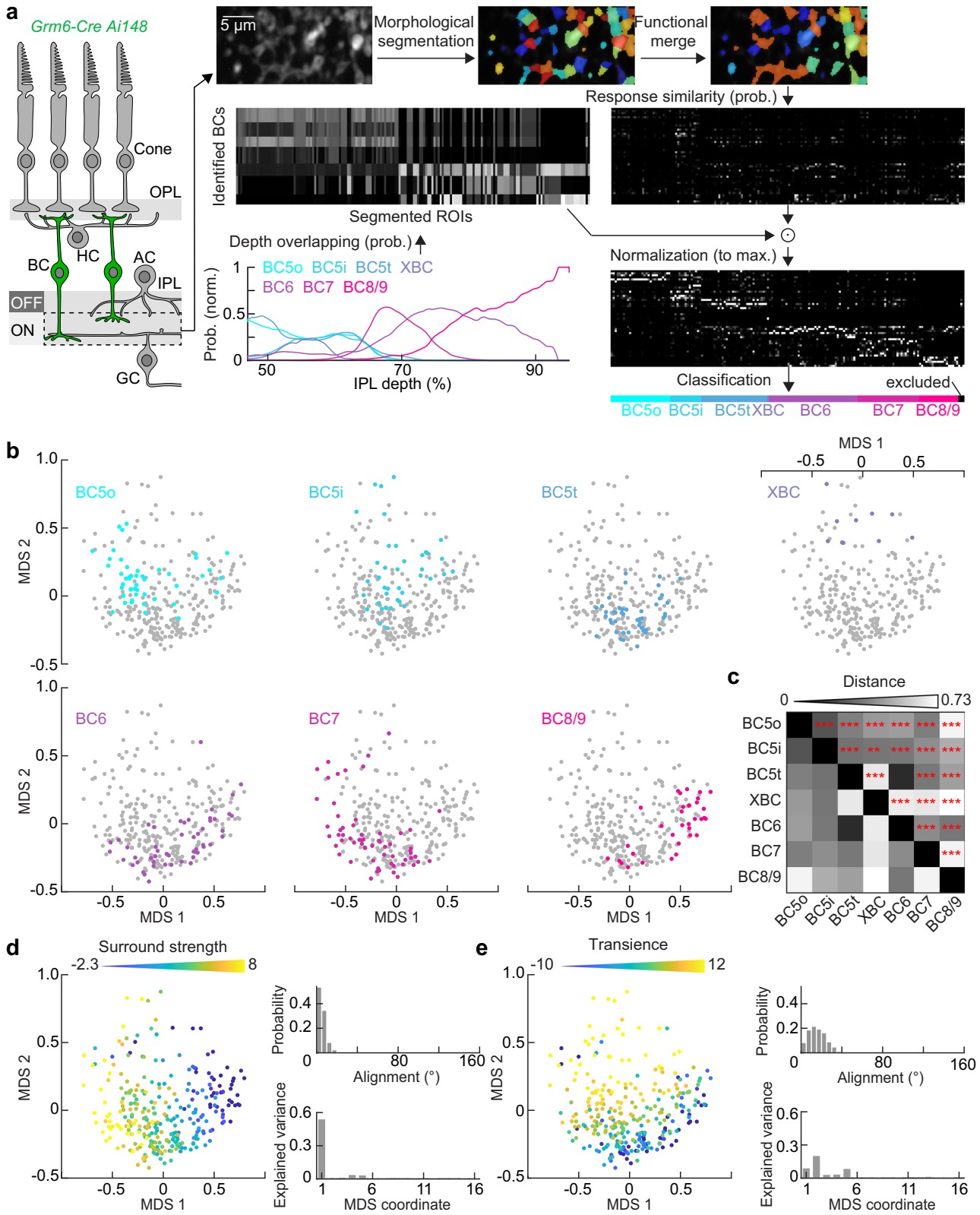

We showed two to four repeats of 11 representative clips (9.8–11.5 s) from these movies to each bipolar cell (Fig. 5a). For all cell types, the luminance within the receptive field center (200 ms before the present) accounted for a majority of the response variance within the limits of repeat reliability (Fig. 5b). Despite this shared encoding of local luminance, ON bipolar cell types differed significantly in their

responses to naturalistic stimuli. This was evident from visual inspection of their average response traces (Fig. 5a) and statistical analyses of their correlation distances (Fig. 5c).

We mapped the encoding space of ON bipolar cells for naturalistic stimuli using a nonmetric multidimensional scaling analysis (MDS, Fig. 5d). Similar to our analysis of a simple artificial stimulus (i.e., spots

**Fig. 4 | Feature encoding from artificial stimuli across ON bipolar cells.**
**a** Schematic representation of the data augmentation process for classifying additional ROIs from calcium images of ON bipolar cell terminals, acquired from *Grm6-Cre Ai148* mice. This classification leverages morphologically identified bipolar cells in conjunction with the scanning depth of the IPL. A semi-manual morphological segmentation approach is utilized, relying on the standard deviation of each pixel over time. To ensure diverse sampling, optimal k-means clustering is employed to group functionally similar segments. Classification hinges on the closest functional similarity and IPL profile match from the identified bipolar cell types. Following this, depth and functional probability maps are arranged based on the classification of the seven distinct bipolar cell types. Outliers are highlighted in black and excluded (<2% of ROIs, see Methods). **b** The encoding space, informed by both functional and morphological data, uses the first two coordinates from nonmetric multidimensional scaling (MDS1 and MDS2) for representation. The x and y axes are set to these coordinates. Individual bipolar cell types are differentiated by color-coding: BC5o ($n = 45$), BC5i ($n = 40$), BC5t ($n = 45$), XBC ($n = 10$), BC6 ($n = 61$), BC7 ($n = 54$), BC8/9 ($n = 35$). **c** Presents the distance matrix (the distance in the first three primary coordinates of the encoding space for artificial stimuli) for each bipolar cell type pairing. Asterisks in the upper triangle indicate statistical significance (* $p < 0.05$, ** $p < 0.01$, *** $p < 0.001$). Two-sided permutation tests were performed, $p$-values have been adjusted for multiple testing by controlling the false discovery rate (FDR) at a threshold of <0.05. **d** Left: The encoding space from (**b**) is overlaid with each ROI's surround strength. Gradient colors encode the value of the surround strength. Top right: A probability distribution captures the angular relationships between subsampled paired vectors (90 pairs or 0.21%) and the collective paired vector angles for surround strength (as detailed in Methods). Bottom right: Each MDS coordinate's variance contribution to surround strength is charted. **e** This panel parallels (**d**) but for response transience. Source data for this figure are provided as a Source Data file.

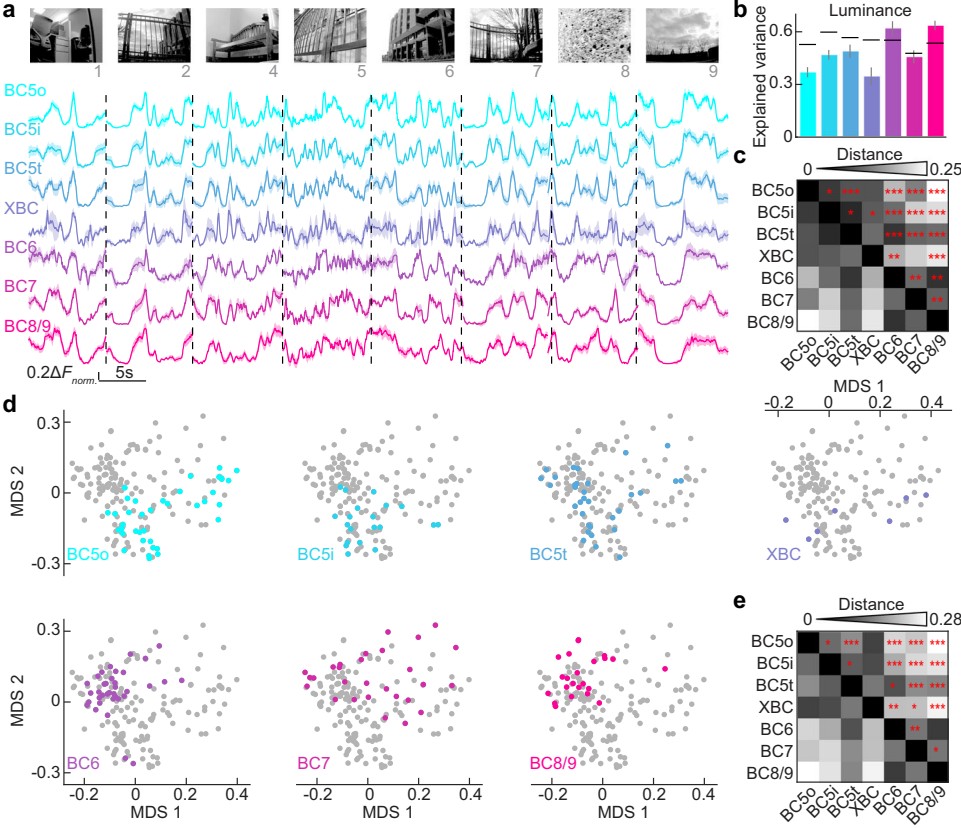

**Fig. 5 | ON bipolar cell responses and encoding space for naturalistic stimuli.**
**a** Color-coded traces of the responses (mean ± SEM) of ON cone bipolar cell types to naturalistic movie clips ranging from 9.8 to 11.5 s in duration. Sample sizes as follows: BC5o ($n = 38$), BC5i ($n = 20$), BC5t ($n = 33$), XBC ($n = 7$), BC6 ($n = 36$), BC7 ($n = 28$), BC8/9 ($n = 23$). Responses to eight of 11 movie clips are shown, as identified below the representative frames. **b** The variance of bipolar cell responses to naturalistic stimuli is dominated by local luminance. Bar charts depict the mean, with error bars representing ±SEM. Sample sizes as follows: BC5o ($n = 31$), BC5i ($n = 16$), BC5t ($n = 21$), XBC ($n = 7$), BC6 ($n = 21$), BC7 ($n = 20$), BC8/9 ($n = 18$). Not all ROIs in (**a**) were repeated at least twice. The repeat reliability, visualized as a black line, is derived from the squared correlation coefficient between two repeated response sets. This can be less than the explained variance, which is based on the squared correlation coefficient between averaged responses from repeats and the preceding 200 ms luminance trace. The explained variance and repeat reliability of each individual data point in the samples can be found in the Source Data. **c** Distance matrix with distance measured as $1 - correlation\ coefficient$ between each bipolar cell type pair. Statistical significance is marked in the upper triangle by asterisks (* $p < 0.05$, ** $p < 0.01$, *** $p < 0.001$), with sample sizes consistent with (**a**). Importantly, ROI classification was based on responses to spot stimuli, not naturalistic stimuli to avoid circularity. **d** The encoding space, calculated by response variances to naturalistic stimuli, is presented with the first two coordinates of nonmetric multidimensional scaling (MDS1 and MDS2) as defined in the x and y axes. Distinct bipolar cell types are demarcated and color-coded, maintaining the counts from (**a**). **e** The distance measure in the first three primary coordinates of encoding space between each bipolar cell type pair. The upper triangle denotes statistical significance (* $p < 0.05$, ** $p < 0.01$, *** $p < 0.001$) with sample sizes matching those in (**a**). For (**c**, **e**), two-sided permutation tests were performed, and their $p$-values have been adjusted for multiple testing by controlling the false discovery rate (FDR) at a threshold of <0.05. Source data for this figure are provided as a Source Data file.

of varying size), the first two principal coordinates (MDS1 and 2) dominated the variation between the responses of ON bipolar cells (Supplementary Fig. 8). Different ON bipolar cell types, occupied overlapping but distinct areas in the encoding space, indicating that they convey different information about naturalistic stimuli (Fig. 5c). Furthermore, ON bipolar cells in the upper (except for BC5t) and lower block were found in separate sections of the encoding space and seemed to vary along different axes (Fig. 5d).

## Distributed encoding of naturalistic stimulus features across ON bipolar cells

To explore how visual features of natural environments are organized across ON bipolar cells, we calculated several parameters (i.e., local luminance, spatial contrast, temporal contrast, motion coherence, and in-center contrast) of naturalistic stimuli within the receptive field center (and surround) over time and measured the influence of these parameters on the neuronal responses (Fig. 6a). To extract parameters from the correct areas of naturalistic movies, we first mapped the receptive field of each cell by reverse correlating its responses to a binary checkerboard white noise stimulus (Supplementary Fig. 12)[57]. We then confirmed that slight variations in the receptive field positions from the center of the stimulus display in our data (95% of ON bipolar cells had receptive field centers within 18.3 μm from the display center) did not meaningfully affect the parameters extracted from naturalistic movies (Supplementary Fig. 13).

When we overlayed the feature sensitivities for naturalistic movies on the encoding space of ON bipolar cells, we found that spatial contrast (i.e., luminance differences between the receptive field center and surround) and temporal contrast (i.e., temporal luminance changes in the center and surround) accounted for differences between ON bipolar cell responses (Fig. 6b, c). Spatial and temporal contrast

sensitivity for naturalistic stimuli are analogous to surround strength and transience, respectively, measured for responses to spots of varying size (Fig. 4). ON bipolar cell axons in the upper block diverge in their sensitivities to temporal contrast of naturalistic movies (Supplementary Fig. 14), whereas ON bipolar cell axons in the lower block differed in their sensitivities to spatial contrast (Supplementary Fig. 14).

Differences in spatial contrast sensitivity dominated the first principal coordinate of the encoding space (Fig. 6b), whereas temporal contrast sensitivity was distributed across the first three principal coordinates (Fig. 6c), highlighting the independent distributed encoding of spatial and temporal information across bipolar cells (Supplementary Fig. 10). In addition, sensitivity to coherent motion through the receptive field contributed to variation in the first and second principal coordinates (Fig. 6d). The angle of variation in coherent motion sensitivity across the encoding space diverged from that for spatial (35.2°) and temporal contrast sensitivity (20.3°, Supplementary Fig. 10). Consistent with this observation, coherent motion sensitivity varied across bipolar cell types in the upper and lower block (Supplementary Fig. 14, Kruskal-Wallis one-way ANOVA, $p < 0.001$, $\eta^2 = 0.24$). There was little correlation between coherent motion and temporal ($R^2 = 0.00014$) or spatial contrast ($R^2 = 0.023$), indicating that

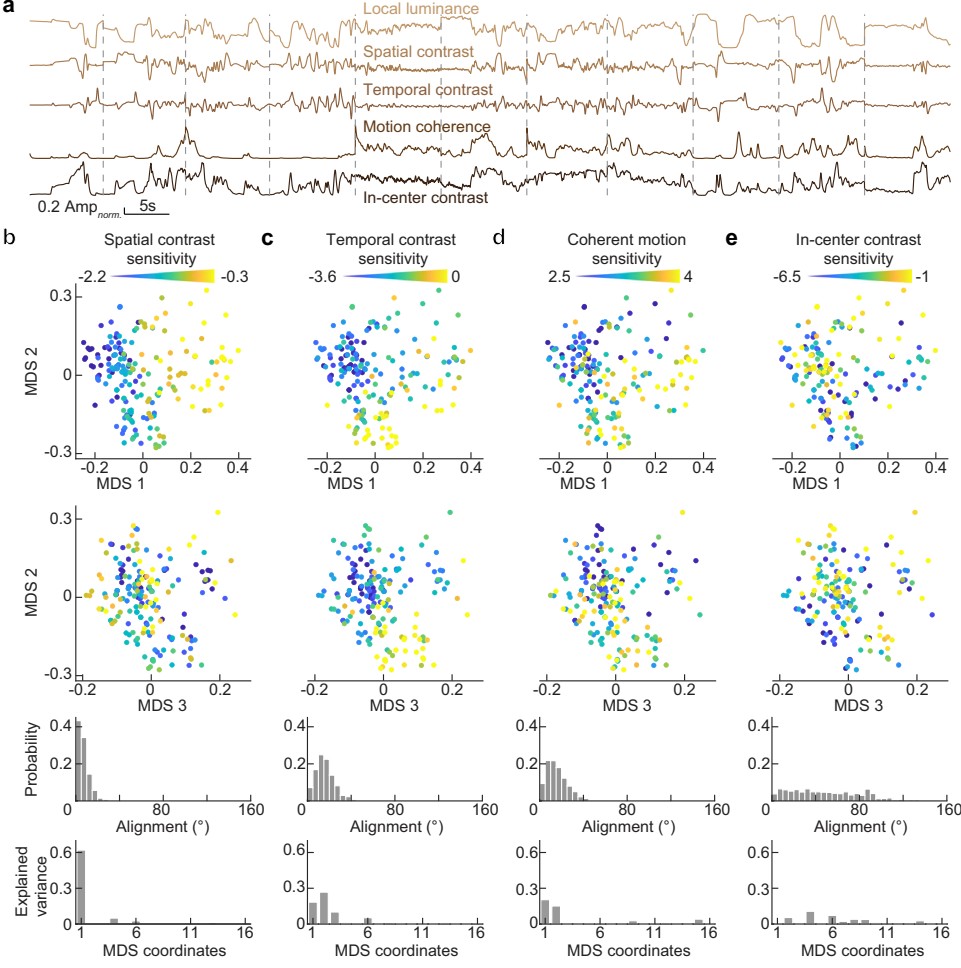

**Fig. 6 | Feature encoding from naturalistic stimuli across ON bipolar cells.**
**a** Traces of five stimulus features, as identified above each trace, derived from 11 movie clips. **b** Spatial contrast sensitivity of each ROI color-coded onto the encoding space (Top: first and second principal coordinates, upper middle: second and third principal coordinates). Lower middle represents the probability distribution of angles, derived from subsampled paired vectors (constituting 90 pairs or 0.53%) in relation to the sum of all paired vectors for spatial contrast sensitivity

(see Methods). Bottom, variance in spatial contrast sensitivity explained by each MDS coordinate. **c** Analogous to (**b**) for temporal contrast sensitivity. **d** Analogous to (**b**) for coherent motion sensitivity. **e** Analogous to (**b**) for in-center contrast sensitivity. The total ROI counts for each panel are as follows: (**b**) $n = 185$, (**c**) $n = 185$, (**d**) $n = 185$, and (**e**) $n = 162$. ROIs with repeat reliability ($R^2$) < 0.2 were omitted from the analysis in (**e**). Source data for this figure are provided as a Source Data file.

they are statistically independent features of naturalistic movies multiplexed in the stimulus encoding of the ON bipolar cell population.

Some bipolar cells in salamanders integrate spatial information within their receptive fields nonlinearly[58]. Nonlinear integration makes bipolar cells sensitive to spatial contrast in their receptive field center (i.e., in-center contrast). We found limited sensitivity to in-center contrast in mouse ON bipolar cells and no significant impact of this parameter on their joint encoding space (Fig. 6e).

Thus, the distributed encoding of spatial and temporal contrast features in complementary blocks of ON bipolar cell types is conserved from artificial to naturalistic stimuli. Motion provides an independent feature of naturalistic stimuli that is distributed across ON bipolar cells in both blocks.

### Homogeneous processing of naturalistic stimuli in individual bipolar cell axons

Having analyzed the distribution of visual information across parallel pathways composed of different ON bipolar cell types, we next wanted to test whether the axon arbors of individual bipolar cells respond to naturalistic stimuli homogenously or heterogeneously to understand if subcellular processing further diversifies and organizes visual information of bipolar cells in the inner retina or if their axon arbors can be considered computational units. We anatomically segmented axon arbors into ROIs isolating presynaptic boutons (Fig. 7a). Figure 7b shows representative responses of three ROIs of individual axon arbors to two repeats of four naturalistic movie clips for all ON bipolar cell types. Pairwise analyses of ROIs in the same arbor revealed that, within stimulus repeats, responses were highly correlated (i.e., overall high correlation coefficient of orange dots in Fig. 7c, d). ROI pairs with lower correlation coefficients invariably had lower repeat reliability in their stimulus responses, indicating that the respective traces were shaped by noise or spontaneous activity (i.e., most orange dots are confined to the upper left quadrants in Fig. 7c). Similarly, comparing the responses of ROI pairs to different stimulus repeats showed that correlation coefficients closely followed repeat reliability (i.e., purple dots in Fig. 7c cluster around the unity diagonal). Together these results suggest that light responses are uniform across axon arbors and that heterogeneities in axonal calcium transients result from noise or spontaneous activity. This finding held for all ON bipolar cell types.

True heterogeneity of light responses within axon arbors would be expected to increase with the distance between the ROIs examined. We observed no distance dependence in the correlation coefficients of ROI pairs, neither within stimulus repeats nor across (Fig. 7d). Finally, we estimated the length constants of signal integration in ON bipolar cell axons by modeling the expected pairwise correlations and comparing them to our observations (Fig. 7e). For all ON bipolar cell types, the estimated length constants were longer than the equivalent diameter of their axon arbors, independent of whether the ROI distances were measured along the axon path or within the imaging plane (Fig. 7e). Thus, ON bipolar cell axons homogenously process naturalistic stimuli and can be considered computational units.

### Homogenous processing of artificial stimuli in individual bipolar cell axons

We found that bipolar cells distribute spatial and temporal information encoding across complementary blocks of ON types (Figs. 2–6). To test whether spatial or temporal information varies systematically across individual axon arbors, we compared ROI responses to spots of different sizes (Fig. 8). Figure 8a, b show anatomical segmentations and ROI responses of representative bipolar cells of all ON types. Pairwise analyses revealed that neither variations in the ROIs' surround strengths (Fig. 8c) nor response transience (Fig. 8d) depended on the distance between ROIs along the axon path or within the imaging plane. This finding held for all ON bipolar cell types.

Thus, the axons of individual ON bipolar cells do not subcellularly organize spatial or temporal information but respond homogeneously to naturalistic and artificial stimuli.

## Discussion

We examined how retinal ON bipolar cells extract and organize information from artificial and naturalistic stimuli by imaging calcium transients in their axon arbors, which integrate photoreceptor signals conveyed from the dendrites with local input from amacrine cells. Our results support the following main conclusions. (1) ON bipolar cells generate distributed representations of temporal and spatial stimulus features in two blocks of four cell types. (2) The axons of the ON bipolar cell types jointly encoding temporal vs. spatial information separate laminarly in the IPL (upper block: temporal information, lower block: spatial information). (3) ON bipolar cells extract and distribute temporal and spatial features similarly for artificial stimuli and naturalistic movies. (4) For naturalistic movies, ON bipolar cells vary systematically in their responses to coherent motion. (5) Motion information is distributed across the ON bipolar cell types in the upper and the lower block, multiplexed with the encoding of temporal and spatial contrast, statistically independent features of natural environments. (6) The axon arbors of individual ON bipolar cells (of all types) respond homogeneously to artificial and naturalistic stimuli, indicating that they function as computational units.

We reach these conclusions based on functional recordings of morphologically identified bipolar cell types. Function-agnostic classification with a decision tree avoids circularity, geometric biases of clustering algorithms, and oversights from undersampling[59–62]. It allowed us to discover a previously unknown division of the ON pathway into two blocks that distribute temporal and spatial information encoding, respectively. From our recordings of morphologically identified bipolar cells, we derived a simple classifier (based on function and IPL depth) that allowed us to augment our data with population recordings, leveraging their higher throughput to address questions that required more data. This sequential approach, using morphologic (or genetic) identification of cell types for targeted recordings of individual neurons to derive a functional classifier for analyses of large-scale population recordings, could help analyze information processing and distribution in other parts of the nervous system.

Analyses of published serial EM reconstructions[8,12,40] and our light microscopy data revealed that ON bipolar cell axons stratify in two blocks of four types (Fig. 1). The upper block comprises BC5o, BC5i, BC5t, and XBC, and the lower block BC6, BC7, BC8, and BC9. Axons within each block overlap extensively and likely converge on postsynaptic targets, whereas axons in different blocks likely innervate different partners. It has been reported that bipolar cell axons stratifying toward the center of the IPL respond more transiently to light steps than axons stratifying toward the edges[6,11,43–47]. Our data recapitulate this overall trend (Fig. 3). More importantly, they reveal that temporal tuning is diverse within the upper block and uniform in the lower block of ON bipolar cells. In the upper block, XBC has higher and BC5t lower frequency cut-offs than the other bipolar cell types (Fig. 2), whereas all types in the lower block filter temporal frequencies with similar cut-offs (Fig. 2). Responses to light steps range from transient (XBC) to sustained (BC5t) in the upper block, whereas all types in the lower block exhibit sustained responses (Fig. 3). Finally, MDS analyses revealed that ON bipolar cells in the upper but not in the lower block distribute the encoding of temporal information extracted from artificial and naturalistic stimuli across types (Figs. 4, 6, 7). Ichinose et al. characterized the temporal tunings of ON bipolar cells by patch-clamp recordings[23]. Their findings matched ours for some types (e.g., XBC) but not for others (e.g., BC7)[23]. Ichinose et al. recorded somatic voltages in the presence of inhibitory synaptic blockers to isolate the dendritic processing of photoreceptor signals[23]. Thus, consistencies in

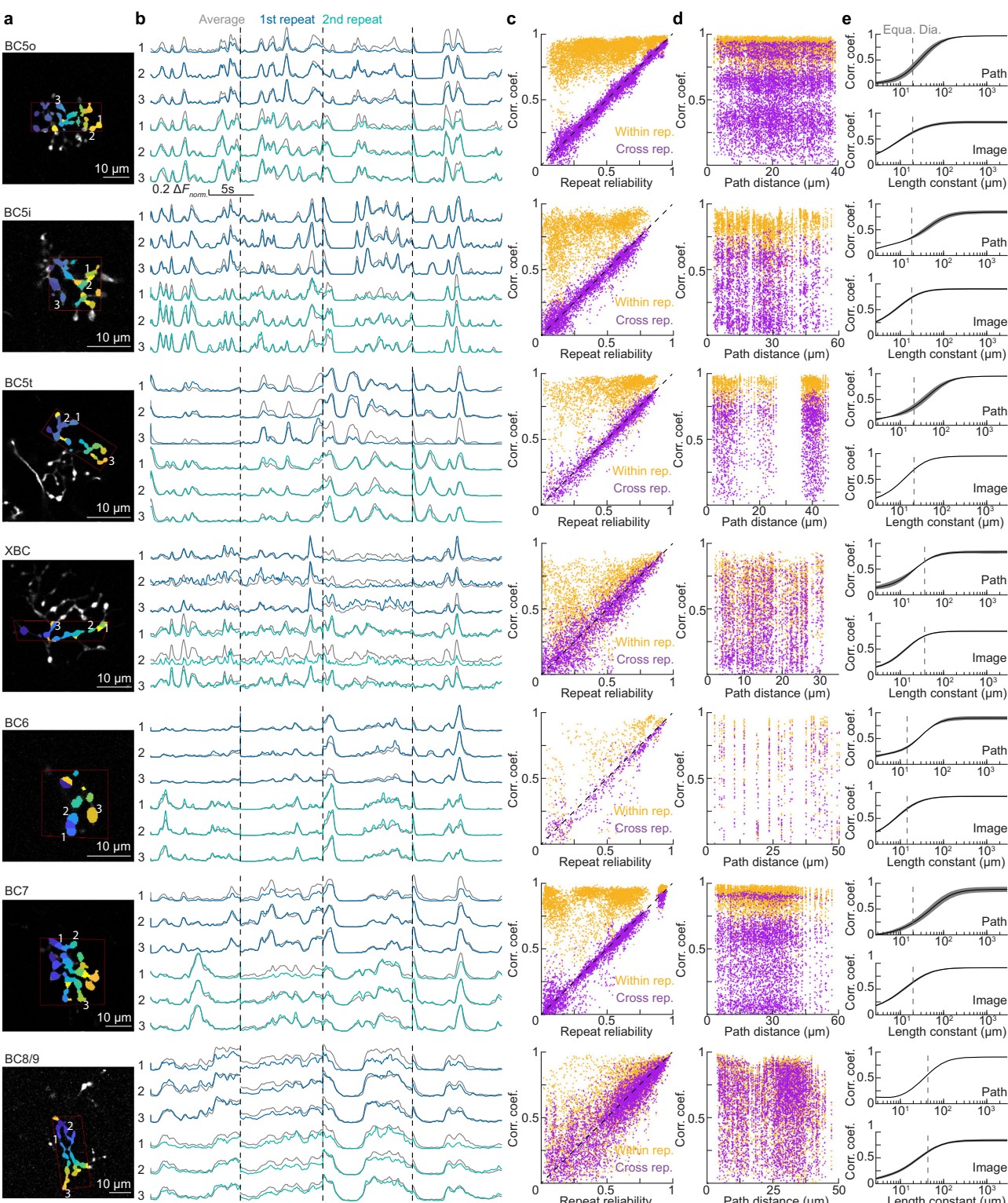

our findings suggest that dendritic processing of photoreceptor inputs dominates the axonal responses of the respective bipolar cell types. In contrast, discrepancies likely reflect the local influence of amacrine cells on others.

Few previous studies analyzed the spatial tuning of ON bipolar cells, with varying results. Some reported homogenous surround strengths[11,63], whereas others observed diversity but either focused on individual cell types[64] or combined the results of many[24]. All studies so far have been restricted to artificial stimuli. We analyzed the spatial processing of artificial and naturalistic stimuli across all ON bipolar

cells. We found that surround strengths were uniform across cell types in the upper block but varied widely between cell types in the lower block (Fig. 3). Furthermore, MDS analyses revealed that ON bipolar cells in the lower but not in the upper block distribute the encoding of spatial information extracted from artificial and naturalistic stimuli across types (Figs. 4, 6, 7). Thus, we uncover a new organizing principle of the ON pathway, its division into two laminar blocks in the IPL that distribute the encoding of temporal and spatial information across their constituent cell types. It will be interesting to test whether this organization is conserved across mammals, like the laminar separation

**Fig. 7 | Subcellular responses of ON bipolar cells to naturalistic stimuli.**
**a** Representative images of segmented bipolar axonal arbors of all cell types. Functional imaging plane outlined in red on top of the anatomical z-stack for path distance measurement based on 3D skeleton reconstruction. Colors differentiate morphologically separate ROIs. Numbers indicate the three ROIs whose responses are shown in (**b**). 10-µm scale bars are included. **b** Response traces from the three representative ROIs in (**a**) to naturalistic stimuli. Two repeats (first: blue; second: green) of four representative movie clips for the individual ROIs and the average off all ROIs (gray) are shown. **c** Scatter plots of the repeat reliability for each ROI pair vs. their within-repeat (orange) cross-repeat (purple) correlation coefficients. Calculations were performed on a per-movie-clip basis. **d** Scatter plots of the path distance between ROIs from 3D-reconstructed axon arbors vs. their within-repeat

(orange) cross-repeat (purple) correlation coefficients. **e** Top: The correlation coefficient between observed data and projected data, charted as a function of the length constant reliant on path distances measured in 3D reconstructions (see Methods); the traces (shaded areas) indicate the mean (±SEM) of the correlation coefficient. Dashed vertical lines indicate the equivalent diameter of axon arbors of bipolar cells of the respective types. Sample sizes as follows: BC5o ($n = 3$), BC5i ($n = 2$), BC5t ($n = 2$), XBC ($n = 2$), BC6 ($n = 2$), BC7 ($n = 2$), and BC8/9 ($n = 1$). Bottom: mirroring top but using Euclidian distances between ROIs rather than on-path distances (top). Samples sizes as follows: BC5o ($n = 11$), BC5i ($n = 5$), BC5t ($n = 2$), XBC ($n = 4$), BC6 ($n = 5$), BC7 ($n = 7$), and BC8/9 ($n = 4$). Source data for this figure are provided as a Source Data file.

of ON and OFF pathways[6], and if the OFF pathway mirrors the ON pathway's division of temporal and spatial information. Anatomically, the OFF pathway shows a similar separation into two blocks of axons (upper: BC1A, BC1B, and BC2; lower: BC3A, BC3B, and BC4)[8,40,65].

In each ON pathway block, two cell types showed similar spatio-temporal feature preferences: BC5i and BC5o in the upper block and BC8 and BC9 (which we combined in our analyses) in the lower block. Interestingly, BC5i axons were recently shown to signal the orientation of visual stimuli, preferring vertical over horizontal edges[26]. Similarly, BC9 selectively contact true S-cones throughout the retina to convey chromatic information[31,38]. Thus, orthogonal features are added to the distributed encoding of temporal and spatial luminance contrast in the two ON pathway divisions. Why orientation-selective signals are grouped with the temporal channel and color-selective signals with the spatial channel remains to be explored.

We find that ON bipolar cell responses to naturalistic stimuli are dominated by local luminance (Fig. 5) and that ON bipolar cells similarly distribute the encoding of spatial and temporal contrast information for artificial and naturalistic stimuli across cell types (Figs. 2–6). Thus, the inner and outer retinal circuits that control bipolar cell activity appear to function similarly under artificial and naturalistic stimulus conditions. Because of increasingly complex neural interactions, the same does not hold for ganglion cells, postsynaptic targets of bipolar cells and the output neurons of the eye[66,67].

In addition to luminance (shared), spatial and temporal contrast (distributed across the upper and lower block, respectively), ON bipolar cells extract motion information from naturalistic stimuli (Fig. 6). Sensitivity to coherent motion is distributed across cell types in both blocks, multiplexed with differences in temporal contrast sensitivity (Figs. 5, 6). Motion sensitivity is higher in the upper block (Supplementary Fig. 9), which aligns with the dendritic stratification of ganglion cells that encode different speeds and trajectories of object- and self-motion[1,68–70]. In the salamander retina, some bipolar cells integrate spatial information nonlinearly, driving responses to contrast within their receptive field center[58]. In mice, we find no evidence for nonlinear spatial integration of naturalistic stimuli among ON bipolar cells (Fig. 6). Thus, the sensitivity to spatial contrast in the mouse retinal output (i.e., the receptive field subunits of ganglion cells[71–73]) is likely limited by the size of bipolar cell receptive fields.

Another crucial organizational question is whether the axon arbors of individual ON bipolar cells are divided into functionally distinct regions or operate as computational units. Subcellular processing and differential distributions of visual information are common in neurite arbors of amacrine cells[32,36,74–76]. Recently it has been suggested that the same may apply to the axon arbors of bipolar cells, which could greatly expand the number of parallel pathways in the inner retina[25,42,77]. However, other studies indicated that bipolar cell axons process visual information homogeneously[11,64,78,79]. We surveyed how axon arbors of all ON bipolar cells integrate artificial and naturalistic stimuli (Figs. 7, 8). We find no evidence for local processing in comparing responses across axon arbors, and our modeling indicates that the length constants of integration exceed the equivalent diameter of

axon arbors. These findings held for all ON bipolar cell types. Thus, we conclude that bipolar cell axons function as computational units. However, it remains possible that the output of axon arbors is diversified by local synaptic or neuromodulatory influences on the release machinery[79].

Our insights into the organization of visual information about natural environments with assignments to specific morphologically identified ON bipolar cell types, in conjunction with progress in retinal connectomics[8,12,40,65,80–82], should advance biologically plausible/realistic circuit models for understanding retinal computations, a long-standing goal of vision science[6,83].

## Methods
### Animals
Throughout this study, we used mice expressing the genetically encoded calcium indicator GCaMP6f in a Cre-dependent tTA-amplified manner (*Ai148* mice)[35,36]. We subretinally injected *Ai148* mice with *AAV-Grm6-Cre* or crossed them to *Grm6-Cre* mice[52] for sparse and dense ON bipolar cell targeting, respectively. For subretinal AAV injections, mouse pups (postnatal days four to six) were anesthetized on ice, and 200 nL *AAV-Grm6-Cre* was delivered into the subretinal space with a Nanoject II injector (Drummond). For two-photon imagining, we isolated retinas from adult mice of either sex. Data from morphologically identified ON bipolar cells included in this study were obtained from 41 mice (22 females, 19 males) with an average age of $2.9 \pm 0.6$ (standard deviation) months. We observed no sex-specific differences in our results and, therefore, combined data from males and females. Mice were housed on a 12-h light/12-h dark cycle. Behavioral experiments were conducted at 20–21 °C with 30–50% humidity levels. All procedures in this study were approved by the Animal Studies Committee of Washington University School of Medicine (Protocol # 23-0116) and performed in accordance with the National Institutes of Health *Guide for the Care and Use of Laboratory Animals*.

### Adeno-associated viruses
Viral particles were packaged and purified as previously described in refs. 84,85. Briefly, AAV1/2 chimeric virions, which readily infect bipolar cells[86], were produced by co-transfecting HEK293 cells with *pAAV-Grm6-Cre*, in which four repeats of a 200 bp *Grm6* promoter element[87,88] drive expression of Cre recombinase, and helper plasmids encoding Rep2 and the Caps for serotypes 1 and 2. Forty-eight hours post-transfection, we collected the cells and their surrounding fluid. We then purified viral particles via heparin affinity columns (Sigma).

### Immunohistochemistry
After two-photon calcium imaging, the retinas were transferred to membrane disks (HABGO1300, Millipore) and fixed for 30 min with 4% paraformaldehyde in mACSF$_{HEPES}$ containing (in mM): 119 NaCl, 2.5 KCl, 1.3 MgCl$_2$, 1 NaH$_2$PO$_4$, 2.5 CaCl$_2$, 11 glucose, and 20 HEPES, with pH adjusted to 7.37 using NaOH. To prepare for immunostaining, retinas were cryoprotected in 10% sucrose in PBS for 1 hr at RT, 20% sucrose in PBS for 1 hr at RT, and 30% sucrose in PBS overnight at 4 °C. We then

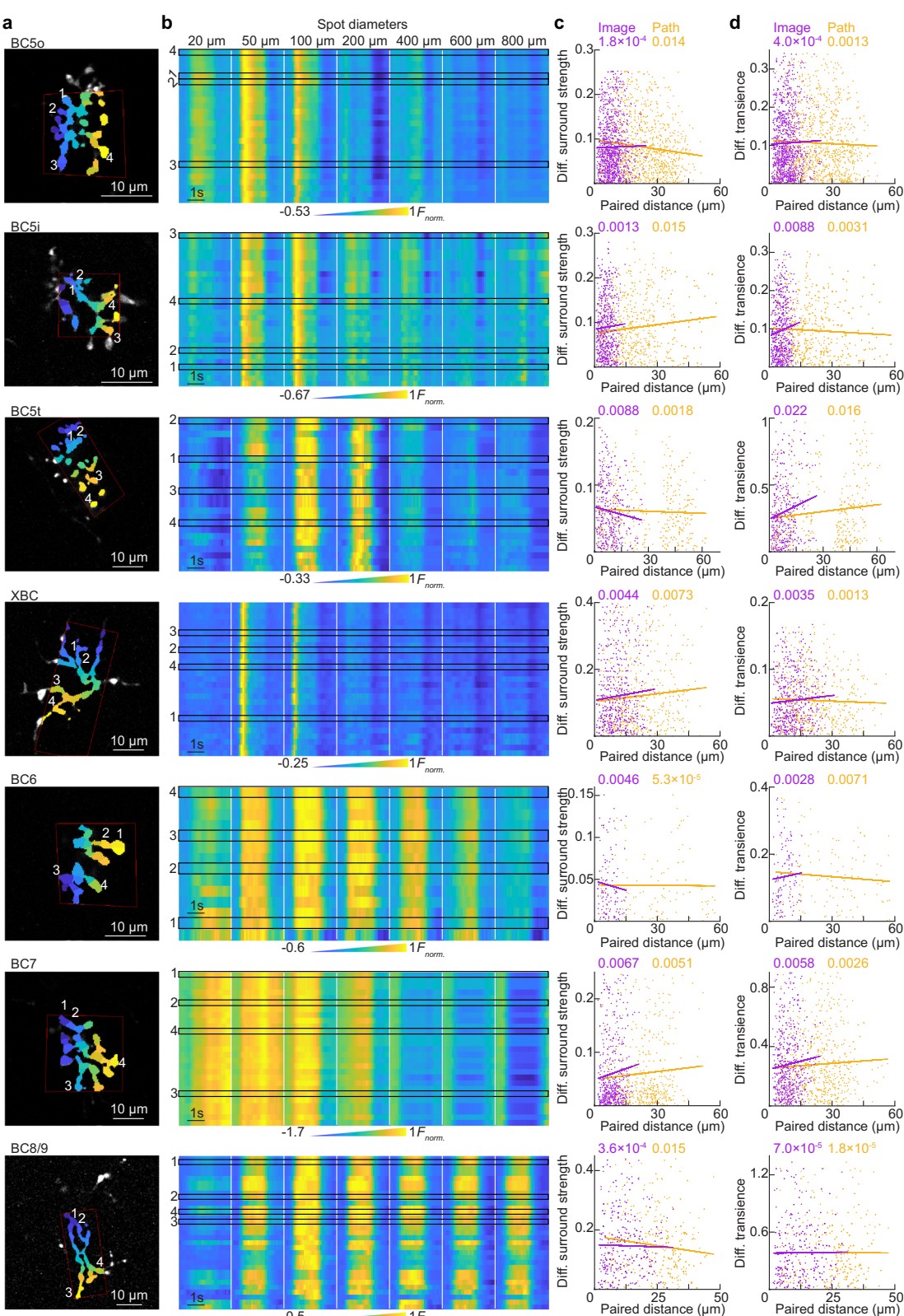

freeze-thawed the retinas three times to enhance permeability, after which they were washed three times for 10 min each in PBS, blocked with 10% normal donkey serum in PBS for 2 h at RT, and incubated with primary antibodies for 5 days at 4 °C. We used the following primary antibodies in this study: chicken anti-GFP (1:1000, ThermoFisher, RRID:AB_2534023), rabbit anti-cone arrestin (CAR, 1:1000, Millipore, RRID:AB_1163387), and goat anti-choline acetyltransferase (ChAT,

1:200, Millipore, RRID:AB_11213095). After primary antibody incubation, we washed retinas three times for 10 min each in PBS at RT and stained with secondary antibodies conjugated with Alexa 488 (1:1000, anti-chicken IgY, ThermoFisher, RRID:AB_2534096), Alexa 568 (1:1000, anti-rabbit IgG, ThermoFisher, RRID:AB_2534017), and Alexa 633 (1:1000, anti-goat IgG, ThermoFisher, RRID:AB_2535739) overnight at 4 °C. Following another three 10-min PBS washes, the retinas were

**Fig. 8 | Subcellular responses of ON bipolar cells to artificial stimuli.**
**a** Representative images of segmented bipolar axonal arbors of all cell types. Colors differentiate morphologically separate ROIs. Numbers indicate the four ROIs whose responses are highlighted in (**b**). 10-μm scale bars are included. **b** Heatmaps or the responses of all ROIs (rows) of representative ON bipolar cells to spots of varying size (columns). Outlines highlight the four ROIs indicated in (**a**). **c** Scatter plots of charting the differences in surround strength between ROI pairs vs. their on-path (orange) separation measure from 3D reconstructions of axon arbors or Euclidian distances (purple) in the image stack. Data pooled from all 3D reconstructed

bipolar cells. Cell counts: BC5o: $n = 3$, BC5i: $n = 2$, BC5t: $n = 2$, XBC: $n = 3$, BC6: $n = 2$, BC7: $n = 2$, BC8/9: $n = 1$. Data points (i.e., ROI pairs; shared for both distance metrics): BC5o: $n = 1026$, BC5i: $n = 681$, BC5t: $n = 414$, XBC: $n = 505$, BC6: $n = 121$, BC7: $n = 481$, BC8/9: $n = 435$. Linear regression lines are plotted for each set, with the $R^2$ values given at top in matching colors. **d** Analogous to (**c**) for differences in response transience. Cell counts same as in (**c**), data points (i.e., ROI pair) as follows: BC5o: $n = 1030$, BC5i: $n = 694$, BC5t: $n = 413$, XBC: $n = 539$, BC6: $n = 126$, BC7: $n = 590$, BC8/9: $n = 421$. Source data for this figure are provided as a Source Data file.

---

mounted in Vectashield medium (Vector Laboratories, RRID:AB_2336789).

### Confocal imaging and bipolar cell classification

Confocal images of bipolar cells were acquired on an FV1000 laser scanning microscope (Olympus) with a 60 × 1.35 NA oil immersion objective (Olympus). Image stacks were captured at a voxel size of 0.066–0.3 μm (x/y – z axes). For the anatomical analysis, image stacks were projected using Fiji → Image → Stacks → Z Project, then axon and dendrite territories were measured using Fiji → Analyze → Measure. For stratification profiles in the IPL, side view of image stacks was first processed using Fiji → Image → Stacks → Reslice [/] → Z Project, then fluorescence intensity of axon was measured using Fiji → Analyze → Plot Profile along the IPL distance, whose boundaries were defined by the somas of OFF- and ON- starburst amacrine cells respectively with ChAT staining. For each axon, intensity at each IPL distance / the highest intensity (%) (Y%) was calculated, binned, and plotted with MATLAB (The Mathworks). Bipolar cells were classified into their well-known types by axon territories and stratification profiles in the IPL, and dendritic territories and the number of cone contacts in the outer plexiform layer using a classification tree (Supplementary Fig. 1)[12,18,89,90].

### Stratification analysis

We computed paired correlation coefficients between the IPL profiles of different ON bipolar cell types measured in large-scale EM reconstructions[8] to assess their division into upper and lower blocks, respectively.

### Two-photon imaging

Two-photon imaging was performed on a custom-built upright microscope (Scientifica) controlled via the Scanimage r3.8 toolbox in MATLAB. Data were acquired using a DAQ NI PCI6110 board (National Instruments). The genetically encoded calcium indicator GCaMP6f[91] was excited with a Mai-Tai laser tuned to 930 nm, and its fluorescence emission was collected through a 60×1.0 NA water immersion objective (Olympus). To prevent the visual stimulus light (peak: 385 nm) from hitting the photomultiplier tube, we captured fluorescence through a sequence of filters (1: 450-nm long-pass, Thorlabs; 2: 513–528 nm band-pass, Chroma). Two-photon timeseries were acquired at 9.5 Hz for responses to varying size spots, naturalistic movies, and checkerboard white noise stimuli and at 37.9 Hz for frequency-modulated spots. The average pixel density was 18.4 pixels/μm² (8.19–26.2 pixels/μm2). All recordings were made >500 μm from the injection site to avoid complications associated with potential retinal detachments from subretinal injections. Before each visual stimulus, the retina was adapted to laser scanning for 30 s. All images were acquired in the ventral retina, where S-opsin is dominant[39]. Throughout the experiments, retinas were perfused at a rate of ~6 mL/min with 33 °C mACSF$_{NaHCO3}$, containing (in mM) 125 NaCl, 2.5 KCl, 1 MgCl$_2$, 1.25 NaH$_2$PO$_4$, 2 CaCl$_2$, 20 glucose, 26 NaHCO$_3$, and 0.5 L-glutamine, equilibrated with a 95% O$_2$ and 5% CO$_2$ mixture.

### Image preprocessing

During two-photon imaging, we simultaneously acquired transmitted light and fluorescence signals to detect z-axis displacements. Image

series that exhibited such displacements were excluded from further analyses. For the remaining series without z-axis drift, images were registered using MATLAB's built-in image registration function, restricted to rigid translation between frames. The translation matrix was estimated based on the transmitted light images and applied to both the transmitted light and fluorescence images. Following successful registration, we applied a 3D median filter (3 × 3 × 3-pixel kernel) to the fluorescence signal to counteract acquisition noise. An initial threshold was set for pixel inclusion in subsequent analyses to capture the top 75% of pixels in standard deviation over the recording time. This automated thresholding was refined by manual visual inspections, aiming to isolate signals from bipolar cell terminals and remove background noise. Responses from eligible pixels were epoched by each stimulus condition. For varied-size spots and frequency-modulated spot stimuli, the average response from 0.5 s before stimulus onset was subtracted as a baseline correction.

### Morphological ROI segmentation

We employed a semi-automated process based on the temporal standard deviation of the calcium images to morphologically segment bipolar cell images into ROIs. Segmentation involved the following steps: First, local peaks were identified using a threshold. Initially, this threshold is set at the 75th percentile of all pixel values. It was then fine-tuned based on visual inspection. Second, ROIs are grown from local peaks, assigning each above-threshold pixel to its nearest peak while ensuring a continuous connection. Third, if visual inspection indicated that further segmentation was necessary, new ROIs were manually added. The center of such a newly added ROI was determined by the peak position among the 12 nearest pixels to the manually assigned ROI location. Once added, the segmentation process was recalculated to integrate the new ROI. Fourth, after the segmentation, each pixel in the 2D image—having collapsed the time dimension—was assigned an ROI number based on the segmentation. Pixels not included in any ROI were set to zero, excluding them from further analyses.

### Visual stimulation

Visual stimuli generated in MATLAB were presented via the Cogent Graphics toolbox developed by John Romaya at the Laboratory of Neurobiology at the Wellcome Department of Imaging Neuroscience from a UV E4500 MKII PLUS II projector illuminated by 385-nm LED (EKB Technologies). Neutral density filters (Thorlabs) were used to attenuate the projector's output, which was focused onto the photoreceptors via the substage condenser of the upright two-photon microscope (Scientifica). Stimuli were shown in 800 μm diameter circular areas on the ventral retina. Before each visual stimulus, we adapted the retina for 30 s to laser scanning and the average background intensity of the stimulus. We presented spots of seven different sizes (diameters: 20, 50, 100, 200, 400, 600, and 800 μm, i.e., varying size spot stimulus) in a pseudorandom sequence. Each spot turned ON (4.04 × 10³ isomerization/cone/second or R*) for 1.5 s and OFF for 1.5 s (1.02 R*). The ON-OFF sequence was repeated five times in a row for each spot size, with 1.5-s gaps (2.02 × 10³R*) between sizes. To test temporal frequency responses, the intensity in spots of two diameters (150 and 800 μm) was sinusoidally for 2 s six at different frequencies (0.5, 1, 2, 4, 8, and 16 Hz). The sinusoidal modulation spanned the

projector's full range ($1.02$–$4.04 \times 10^3$ R*). Different stimulus frequencies were shown in pseudorandom sequences (each frequency appearing five times) with 1.5 s gaps ($2.02 \times 10^3$ R*) between stimuli. The scan region for temporal frequency responses was half the size used for other stimuli to increase the signal-to-noise ratio for each ROI. At a scan rate of 37.9 Hz, we could discern signal fluctuations up to 18.95 Hz, given the Nyquist criterion. For naturalistic movies, we presented 11 clips at 52.5 Hz, each lasting 9.8–11.5 s (average: $11.22 \pm 0.55$ s) and ranging in average intensity from $1.30 \times 10^3$–$2.55 \times 10^3$ R*. The acquisition of these clips is described in the following section. Movie clips were separated by gaps of $1.43 \pm 0.29$ s. For checkerboard white noise stimuli, each frame consisted of a $15 \times 20$ grid of $20 \times 20$ μm²-squares. The intensity of each square was determined by a binomial process (OFF squares: 1.02 R*, ON squares: $4.04 \times 10^3$ R*). In total, 900 unique frames were shown in each recording. Their sequence was repeated twice to gauge response quality.

## Acquisition of naturalistic movies

We captured natural scenes on a commercial camera (Crosstour CT9000), recording at a resolution of 1080 p ($1920 \times 1080$ pixels) and a frame rate of 60 frames per second (fps). The camera was secured to a rolling support frame using a flexible gooseneck clamp, positioning it approximately one inch above the ground and aligning it with an azimuth of 60° and an elevation of 30° (Supplementary Fig. 7), simulating the mouse's perspective[54]. We moved the camera at speeds matching to previously reported mouse movement velocities (Supplementary Fig. 7)[55,56]. Following the recording session, we estimated the pacing of each frame and analyzed the distribution of movement speeds. This was done by measuring multiple points along the path using a tape measure and applying shape-preserving piecewise cubic interpolation. To calibrate the camera and assess visual distortion, we employed a method involving the presentation of multiple rings of varying sizes on a sheet of paper. These rings were placed in front of the camera, and the angle and distance of each ring relative to the lens center were measured and compared to the resulting image. To present visual stimuli that mimic natural viewing as closely as possible, we matched the paired pixel distances in the video to visual angles in physical space. We discovered that a 170° fisheye perspective linearized the visual angle in the acquired flat image (Supplementary Fig. 7). We projected stimuli at 1.98 μm/pixel onto flat-mounted retinas. One degree of visual angle covers 32.5 μm on the mouse retina[92]. The projection field of our visual stimulus ($600 \times 800$ pixels) thus covers $37° \times 49°$. All movies were center-clipped according to this calibrated relationship.

## F1 power and phase analyses and GCaMP6 frequency responses correction

Power spectra of responses to frequency-modulated spots were obtained by Fast Fourier Transforms (FFT). Here, $X$ represents the signal in the time domain, and $N$ denotes the length of $X$. We upsampled the signal to 189.4 Hz for enhanced resolution in the frequency domain. The power of each component was calculated as $|FFT(X)/N|^2$. Due to its discrete nature, the power from one or two frequencies nearest to the F1 frequency were combined. The F1 power for each bipolar cell was averaged across stimulus repetitions and normalized to the peak F1 power across all frequencies. Because GCaMP6f acts as a low-pass filter, we corrected the F1 power by subtracting the simulated F1 power of GCaMP6f's responses to identical frequency modulations. We used rise (49.5 ms) and decay (156 ms) time constants derived from in vivo mouse V1 cortex 82 recordings and adjusted for the recording temperature[93] for our simulations. Using these time constants, we constructed a response function and convolved it with the frequency modulation, appending a 1 s baseline value (set at 0.5) to the start and finish for stability. The simulated responses underwent the same analysis as bipolar cell signals. Correction involved subtraction from both

normalized signals. For F1 phase analysis comparing large and small spot stimuli, we derived the phase from the angle of the FFT components. To average the angle over trials and adjacent frequencies, each angle was converted to a vector, followed by calculating the angle of the vector sum. In MATLAB's FFT implementation, −90° aligns with the 0° of the stimuli's single function. We consolidated phases from five frequencies, ranging from 0.5 to 8 Hz, to discern phase differences between spot sizes.

## Assessment of response quality to higher-frequency modulation

To test whether we could reliably analyze responses to frequency-modulated spots against the noise, we applied a simple autoregression model (AR) to capture the temporal correlation of each trial for each ROI and then simulated traces with identical correlation structures not influenced by visual stimulation. The AR model depicts the relationship between consecutive observations as follows:

$$x_t = \phi x_{t-1} + \epsilon_t \tag{1}$$

Where $x_t$ is the response amplitude at time point $t$. $\phi$ is the autocorrelation parameter. $\epsilon_t$ is white noise with a standard deviation matching the distribution of $\{x_1, x_2, \ldots, x_n\}$. To estimate $\epsilon_t$, we applied Ordinary Least Squares estimation:

$$\hat{\beta} = (X^T X)^{-1} X^T Y \tag{2}$$

Where X is the design matrix, each row corresponding to a time points; the first column is a column of ones for the intercept term, and the second column contains the predictor variable data from time $t=1$ to $t=n$ Y is the response variable data from time $t=1$ to $t=n-1$. The estimated parameter $\phi$ is the second element of $\hat{\beta}$.

$$\phi = \hat{\beta}_2 \tag{3}$$

The repeat reliability is measured by the median squared correlation coefficient between the traces of half-split repeated trials, where all possible splits were evaluated.

## Measurement of ROI diameters and corresponding frequency responses

To examine whether ROI size shapes temporal tuning, we segmented ROIs from the recordings of frequency-modulated spots of 42 bipolar cells. The diameter of each ROI was measured manually by drawing a line across the cross-section perpendicular to the axonal branch. To ensure response quality, the ROI size in the image was required to be larger than five square pixels, and repeat reliability had to be >0.1. For comparisons between two ROI-diameter groups, the data were divided by the median diameter. We then averaged the responses across all ROIs within the same size category and estimated the F1 power from those averages. The final value is an average of these F1 power measurements across recordings within a cell. F1 power is corrected for the GCaMP6f temporal dynamics as previously described. For evaluating frequency tuning within a bipolar cell, the analyses are the same, except the ROIs of each recording are split into groups of equal numbers based on their diameter.

## Functional variation between upper and lower blocks

To examine whether sampling from upper or lower blocks provides a wider variation in surround strength or transience, we randomly sampled our data 2,000 times, evenly across all combinations within any given block. Specifically, each time the data were drawn from all combinatory pairs of BC types, and then for each pair, data points were randomly selected from both types. The absolute difference was calculated from each sampling of individual pairs.

### Trial-based peak amplitude analysis across varied-size spots

To investigate the potential for spike activity in ON cone bipolar cells, we tested for a consistent amplitude as the hallmark of regenerative spike events. We evaluated the peak amplitudes trial by trial across spot sizes of 20, 50, 100, 200, and 400 μm in diameter. The time window for detection extended from 0.1 to 1.5 s after stimulus onset, which matched the 1.5 s duration of the stimulus presentation. To ensure response quality, the repeat reliability of five repeated trials for each spot size was required to be higher than 0.5; 81.5% of trials met this criterion. Only the ON phase of the repeated trials was examined. In total, 3215 trials from 49 bipolar cells were analyzed. To detect peaks, we employed MATLAB's built-in function designed to identify peak-like events in the response traces. This same analysis was applied to trial-based responses for both the ON and OFF phases of the stimulus.

### Functional distance metrics

To ensure consistency when comparing functional distances between bipolar cell types across both artificial and naturalistic stimuli, we adopted a correlation coefficient-based measurement. Here, correlation distance was defined as $1 - correlation\ coefficient$ between pairs of response units. We used a Euclidean distance metric when considering the encoding space (i.e., the first three principal coordinates of MDS analyses). We evaluated the significance of distances between specific bipolar cell type pairs with a permutation test, determining whether the observed distance between a pair of bipolar cell types was greater than when the labels of individual cells were interchanged in Monte-Carlo simulations (5000 iterations).

### Augmenting data from individual cells with population recordings

We segmented two-photon imaging series of bipolar cell axon populations in the IPL of *Grm6-Cre Ai148* mice as described under '*Morphological ROI segmentation.*' To eliminate redundancies and avoid over-weighing population recordings, we consolidated functionally similar ROIs recorded in the same imaging plane into a single ROI as follows. First, for every recording column (i.e., multiple IPL depth imaged in the same region of the retina), we compiled responses from all the morphologically segmented ROIs and undertook $k$-means clustering based on their response correlations. To determine the optimal value of $k$, we used the Bayesian Information Criterion (BIC): $BIC = nln\left(\frac{RSS}{n}\right) + kln(n)$, where $n$ stands for the total number of data points, while $RSS$ represents the sum of distances from each point to the centroid of its cluster, specifically when using the correlation distance metric. We then consolidated ROIs within the same $k$-means cluster and the same image plane into single ROIs.

We assigned the consolidated ROIs to specific ON bipolar cell types based on their functional similarity to data obtained from morphologically identified cells and the imaging depth in the IPL. To incorporate the information from the IPL depth, we translated the functional distance into a probability metric using a gamma distribution fit to the functional distances between ROIs and nearest identified cell types. The depth-related probability for each cell type was informed by the skeleton density from large-scale EM reconstructions[8]. For every depth, the probability assigned to each ON cone bipolar cell type was normalized, ensuring the combined probabilities of all types added to one. ROIs were assigned to bipolar cell types based on the highest combined functional and depth probabilities. We applied a gamma function to fit the probability distribution of the combined probability (before normalizing to the maximum) and excluded those are lower than 95% of the data (approximation to two standard deviations in normal distribution), which are less than 2% of total augmented data. Importantly this ROI classification for data augmentation used only the functional distances from responses to varying size spots (i.e., artificial stimuli) and thus avoids circularity for assessing cell type differences in responses to naturalistic stimuli with the augmented data. Correspondences between ROIs across stimuli were determined by registering the location of ROIs within the respective image series.

### Evaluating batch effects via Jensen-Shannon divergence (JSD) for 2D distributions

We used an extensive sub-sampling process to evaluate the potential contributions of batch effects to our data distributions. We generated 2000 combinations of recording set distributions and undertook 20,000 random resamples of all data points. The purpose was to test if the distribution from two subsets of recordings deviates significantly from that of two subsets generated by randomly sampling all data points. We used the JSD for this assessment, targeting the first two principal coordinates of the encoding space. Our null hypothesis was that the batch effect did not shape data distributions in the encoding space and that JSDs between two subsets of recordings, therefore, resemble JSDs between two randomly sampled subsets. The JSD is mathematically defined as:

$$JSD(P\|Q) = \frac{1}{2}D_{KL}(P\|M) + \frac{1}{2}D_{KL}(Q\|M) \tag{4}$$

Where, $M = \frac{1}{2}(P + Q)$ and $D_{KL}$ denotes the Kullback-Leibler divergence. For its computation, we employed a discrete approximation given by:

$$D_{KL}(P\|M) = \sum_{ij} P\left(x_{ij}\right)\log\left(\frac{P(x_{ij})}{M(x_{ij})}\right) \tag{5}$$

To approximate the probability densities $P(x)$, $Q(x)$ and $M(x)$, we utilized a kernel smoothing function specifically targeting bivariate data, primarily the first two MDS coordinates.

### Multidimensional scaling (MDS) analysis

To analyze the feature encoding space of ON bipolar cells, we performed nonmetric MDS analyses on their correlation distances. When examining responses to naturalistic movie clips, any ROIs with a repeat reliability <0.2 (measured as squared correlation coefficient [$R^2$] of the average responses to two splits of repeating stimuli) was omitted. To ensure the integrity of the distance metric, we implemented a weight mask. Any distance deviating by more than three standard deviations from the mean was given zero weight. This impacted 3.52% of the data for naturalistic stimuli and 3.27% for artificial stimuli. We then computed the MDS for 16 dimensions, setting a cap at 2000 iterations using the squared stress criterion.

To measure the variance explained by each MDS coordinate, we calculated a distance metric from the Euclidean distance of the aggregated coordinates. $R^2$ was then calculated between the response distance metric and the sub-MDS-coordinate distance metric. Importantly, the same binary weight mask was employed to remove outlier values. The outcomes from our 16-dimensional MDS showed $R^2$ values of 0.93 for naturalistic stimuli and 0.98 for artificial stimuli. However, a characteristic of the correlation distance metric meant some pairings did not adhere to the triangle inequality of the Euclidean distance. This inconsistency was observed in 1.72% of naturalistic stimuli pairings and 6.34% for artificial stimuli. Collectively, the first three principal coordinates accounted for >0.59 of the explained variances.

We chose MDS over principal component (PCA) or similar analyses for two reasons. First, MDS analyses compute a distance matrix to estimate the dissimilarities between cells based on their complete response traces. It then identifies the dimensions of the space defined by the distance matrix across which the cell population varies the most (i.e., the principal coordinates). Thus, this analysis reveals what organizes the diversity of ON bipolar cells. PCA or similar approaches instead highlight the stimulus dimensions across which responses vary

the most. Thus, it focusses primarily on the stimulus space rather than the cellular encoding space.

In addition to this conceptual difference, there is a technical reason for choosing MDS over PCA to analyze responses to naturalistic stimuli. We presented naturalistic stimuli as two to four repeats of 11 representative movie clips. To include a cell's response to a given clip in our analysis, it needed to exceed a quality threshold (i.e., repeat reliability). Because MDS starts by calculating distances between cell pairs, each distance calculation could include all clips for which both cells exceeded the required repeat reliability. Different clips could be included for different cell pairs. In PCA, all cells need to be compared across the same stimulus clips, which would either compromise response quality or drastically reduce the number of admissible clips.

### Assessing the local luminance contributions to responses

To determine the impact of local luminance on bipolar cell responses, we calculated the explained variance ($R^2$) between response traces and local luminance with a 200-ms shift, following the method outlined in the section on *'Spatial contrast sensitivity analysis.'* We restricted our analysis to cells and ROIs for which we had independent repeat recordings from naturalistic movie stimulation. The repeat reliability of each cell and ROI was gauged by the $R^2$ of its averaged responses in two separate subsets of repeated trials.

### Analysis of feature encoding from varying size spots

Responses to varying size spots for each cell or ROI were averaged over all qualifying pixels across repeated trials. Averaged responses were then baseline subtracted and normalized to their peak amplitude before calculating the following feature sensitivities.

Surround strength ($S$)

$$S = \frac{r_c - r_s}{r_c} \tag{6}$$

where $r_c$ represents the average responses to 50 and 100 μm diameter spots centered on the receptive field, and $r_s$ refers to the average responses to 600 and 800 μm diameter spots encompassing the receptive field surround. The response time window for both was 0.34–1.54 s after spot onset.

Response transience ($T$)

$$T = \frac{r_e - r_l}{r_e} \tag{7}$$

where $r_e$ represents the average responses to 50, 100, and 200 μm diameter spots early (0.23–0.67 s) after spot onset, and $r_l$ refers to the average responses to the same stimuli later (1.21–1.65 s after spot onset).

### Analysis of feature encoding from naturalistic movies

Responses to naturalistic movies for each cell or ROI were averaged over all qualifying pixels across repeated trials before calculating the following feature sensitivities.

Spatial contrast sensitivity ($S$)

$$S = \log\left(\frac{Corr(g(S_c - S_s), R)^2}{Corr(g(S_c), R)^2}\right) \tag{8}$$

where $R$ denotes the response of the cell or ROI to the naturalistic movie, $S_c$ represents the contrast in a 100 μm diameter receptive field center, and $S_s$ corresponds to the contrast in a 600-μm diameter area encompassing the receptive field surround. The function $g$ convolves the contrast traces with a 4 s kernel, with its central component being a 2 s response function of GCaMP6f, described under *'F1 power and phase analyses and GCaMP6 frequency responses correction.'* A

weighted mask was produced to convolve with each movie frame as follows. Inside the designated receptive field, each pixel's weight was determined by normalizing a multivariate normal probability function. Here, sigma was defined as $2\pi r\, l$, where $l$ stands for a 2D identity metric and $r$ signifies the radius in μm. Any weight beyond this receptive field size is set to zero. The mask's weights were normalized by their maximum value. This value for each movie frame was derived from the dot product of the frame pixel values, ranging between -0.5 and 0.5, and the weighted mask vector, all divided by the sum of all weights. *Corr* refers to the correlation coefficient. $g(S_c)$ indicates the local luminance. To account for the response delay of ON bipolar cells, we introduced a 200 ms shift to the contrast traces to achieve temporal alignment. This was equally applied to all other feature calculations.

Temporal contrast sensitivity ($T$)

$$T = \log\left(\frac{Corr(g(f_b(S_c)), R)^2}{Corr(g(f_b(S_c) - f_l(S_c)), R)^2}\right) \tag{9}$$

where $S_c$ represents the contrast in a 100-μm diameter receptive field center. The function $f_b$ convolves the receptive field contrast with a band-pass temporal kernel, a difference of Gaussians approximated by two Gaussian membership functions. These functions are characterized by their standard deviations, means, and weights within a 1 s window. Conversely, the function $f_l$ serves as a low-pass filter and is similarly defined but with different temporal parameters. For $f_b$, the first Gaussian had mean 0.08 s, SD 0.05 s, and weight 1. The second Gaussian had mean 0.12 s, SD 0.12 s, and weight −0.5. For $f_l$, the first Gaussian had mean 0.24 s, SD 0.1 s, and weight 1. The second Gaussian had mean 0.3 s, SD 0.24 s, and weight -0.001.

Coherent motion sensitivity ($M$)

$$M = \log\left(\frac{1}{Corr(g(O_a - O_v), R)^2}\right) \tag{10}$$

where optical flow for each pixel was calculated using MATLAB's built-in *'estimateFlow'* function, analyzing consecutive movie frames, employing the Horn-Schunck method with the smoothness parameter set to three, ensuring a global constraint was in place. The outcome was an optical flow map in which each pixel is represented by a 2D vector. To extract coherent motion in a receptive field, we generated a binary mask setting pixels in a 200 μm area centered on the receptive field to one and the rest to zero. $O_v$ and $O_a$ represent the vector sum and average vector length, respectively, of all pixels within the masked area of the optic flow map.

In-center contrast sensitivity ($I$)

$$I = \log_{10}\left(q(g(S_c), g(S_{cin}), R)\right) \tag{11}$$

where $S_c$ represents the local luminance in a 100 μm diameter receptive field center. $S_{cin}$, denotes in-center contrast calculated by dividing the receptive field center into four quadrants and adjusting the individual contrast using a linear ramp function. We also explored other in-center contrast measures, including the weighted standard deviation of pixel values and increased divisions, up to pixel-wise rectification. We did not observe significant differences between these methods. Function $q$ identifies the optimal ratio $\beta$ that blends center and in-center contrast to achieve the highest correlation with the recorded responses. Here, $\beta$ represents the weight attributed to the in-center contrast, while $1 - \beta$ corresponds to the center contrast.

### White noise-guided receptive field alignment

To determine the receptive field positions of ON cone bipolar cells, which are non-spiking neurons, we correlated their responses with the local luminance ($g(S_c)$) in 80 μm disks spatially weighted with

multivariate normal distributions and temporally convolved with a filter based on the GCaMP6f kinetics[91,93]. To map the receptive fields, we estimated $g(S_c)$ at 441 distinct positions, separated by 4 μm along the x and y axes. We then calculated correlation coefficients between $g(S_c)$ and bipolar cell responses for each of these locations. The position of the receptive field center was then identified as the peak in the resulting map of correlation coefficients, refined using a 2D spline interpolation.

## Spatial correlation of visual features in naturalistic stimuli
We established a grid of distinct receptive field positions extending from −40 to 40 μm along both axes, with 4 μm intervals. This grid facilitated the analysis of each visual feature, with receptive field positions adjusted accordingly. We calculated the correlation coefficient between each receptive field position, as well as the associated distances. For naturalistic movie clips, the correlation coefficient, as a function of receptive field distance, was determined using median values from pairs equidistant within bins of 2 μm, with bin centers ranging from 1 to 76 μm. This methodology was consistently applied to various visual features, including luminance, spatial and temporal contrast, coherent motion, and center-surround contrast.

Additionally, we evaluated the distribution of receptive field positions for each bipolar cell by correlating the responses with simulated responses specific to the receptive field. Recordings with a minimum squared correlation coefficient of 0.1 against simulated GCaMP6f responses were selected, yielding 59 valid recordings. From the estimated receptive field centers, we calculated the standard deviation and the bounds containing 95% of the data to provide a statistical analysis of their spatial distribution.

## Analysis of angular deviation of encoded features
The extraction of visual features of artificial and naturalistic stimuli is described in detail under 'Analysis of feature encoding from varying size spots' and 'Analysis of feature encoding from naturalistic movies,' respectively. To measure the angular deviation in the encoding of these features by ON bipolar cells, we computed a vector $v_{s,f}$ defining the orientation of the encoding in the space defined by the first three principal MDS coordinates as:

$$v_{s,f} = \frac{1}{n}\sum_{i,j}^{n}\left(w_{f,i} - w_{f,j}\right)\left(v_i - v_j\right) \qquad (12)$$

Here, $v_i$ and $v_j$ denote vectors corresponding to paired data points in the encoding space. $w_{f,i}$ and $w_{f,j}$ represent the values of a given feature $f$ for the paired data points. To determine the alignment between an encoding feature and a coordinate, we compute the angle between $v_{s,f}$ and standard coordinate vectors: $v_x = (1,0,0)$, $v_y = (0,1,0)$ or $v_z = (0,0,1)$. The angle, denoted by $\theta$, is calculated as:

$$\theta = \cos^{-1}\left(\frac{v_a \cdot v_b}{|v_a||v_b|}\right) \qquad (13)$$

Where $v_a$ and $v_b$ are placeholders for the vectors under comparison. To measure the angular deviation across ON bipolar cells we calculated the $v_{s,f}$ for 1000 random 90-pair subsamples of the data (for varying size spots this represents 0.21% of the data, for naturalistic movies 0.53% of the data) and then measured the angles between each subsample and $v_{s,f}$ calculated from the complete data set.

## Measuring path distances in 3D bipolar cell reconstruction
We used Fiji[94] to measure the path distance between ROIs in 3D reconstructions of bipolar cells. Bipolar cell axons were skeletonized based on z-stacks acquired by two-photon imaging. Bipolar cell responses to visual stimuli were acquired separately in time series of single optical sections. Time series were registered to the z-stacks

using MATLAB's built-in control-point selection function. The centroid of each ROI was then linked to its nearest point on the skeleton, searched in 0.1 μm increments. We then generated a graph incorporating both ROIs and the skeleton to measure the shortest distance in 3D space between ROIs along the skeleton path.

## Estimating length constants and response heterogeneity in axon arbors
The relationship between the distance separating two ROIs in a bipolar cell axon arbor and differences in their responses can be characterized by a length constant. A shorter length constant indicates lower correlation ($r$) in responses of two ROIs separated by a given distance. We modeled $r$ using an exponential decay formula:

$$r_{d,\lambda} = e^{\left(-\frac{d}{\lambda}\right)} \qquad (14)$$

where $d$ denotes the distance (in μm) between two ROIs, and $\lambda$ represents the axonal length constant (in μm). Notably, the correlation between two ROIs is not solely determined by their separation, but also the quality of their responses. We measured response quality as repeat reliability (i.e., the correlation coefficient between the responses to repeated trials of the same stimulus). This expected paired response consistency of two ROIs ($q_{i,j}$) was computed as:

$$q_{i,j} = \left|Corr\left(R_{i,1},R_{i,2}\right)Corr\left(R_{j,1},R_{j,2}\right)\right| \qquad (15)$$

where indices $i$ and $j$ denote different ROIs. We measured the observed paired reliability of two ROIs ($k_{i,j}$) as:

$$k_{i,j} = \left|Corr\left(R_{i,1},R_{j,2}\right)Corr\left(R_{j,1},R_{i,2}\right)\right| \qquad (16)$$

We next compared the expected and observed response consistencies as a function of the length constant according to:

$$Corr(r_\lambda \odot q, k) \qquad (17)$$

where $\odot$ indicates component-wise multiplication.

## Statistics
Functional data from identified ON bipolar cell types were collected from retinas of 39 mice. Two-photon imaging from an additional eight mice were used for data augmentation. All summary data and response traces are presented as mean ± SEM. Differences between surround strength and transience of the upper and lower blocks of bipolar cell responses were assessed using the Wilcoxon rank sum tests, with results reporting sample sizes, p-values, effect sizes (r), and either z-values or U-values when $n < 30$. Variations within blocks and differences in frequency responses across cell types were analyzed using the Kruskal-Wallis one-way ANOVA, including details such as sample sizes, degrees of freedom, p-values, effect sizes ($\eta^2$), and chi-squared ($\chi^2$) values. The Kolmogorov-Smirnov (K-S) test was used to compare phase shifts in the frequency responses between small and large spot sizes; we report the corresponding sample sizes, p-values and statistic (D), representing effect sizes. Paired-group sample-median comparisons were adjusted using the Tukey-Kramer method to correct for multiple comparisons. Comparisons involving different pairs of bipolar cell types underwent correction for multiple comparisons using the Benjamini-Hochberg procedure. Several permutation tests were conducted, and details on their sampling methods, sample sizes, p-values, and observed effect sizes are detailed bin this Methods section and the respective figure legends. We report exact p-values when they are larger than 0.001; otherwise, we note that $p < 0.001$.

## Software information
ImageJ has been updated during the process, but version 1.53 v was used for most analyses.

MATLAB R2021a was used for data processing and analysis. We designed and presented visual stimuli with MATLAB R2016b and Cogent graphics version 1.33. ScanImage version r3.8 was used for two-photon calcium imaging. Adobe Illustrator (2022, version 26.0.3) was used to assemble the final figures.

## Reporting summary
Further information on research design is available in the Nature Portfolio Reporting Summary linked to this article.

## Data availability
Source data are provided with this paper. All other data are available from the lead contact, Daniel Kerschensteiner (kerschensteinerd@wustl.edu), upon request. Additional data for Fig. 1g is in the supplementary material of https://doi.org/10.1038/nature12346. Source data are provided with this paper.

## Code availability
The custom MATLAB scripts used for analysis are available at https://github.com/Jen-Chun-Hsiang/ONBCEncoding or https://doi.org/10.5281/zenodo.10569732.

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

## Acknowledgements

This work was supported by funding from the National Institutes of Health (EYO26978 and EYO34001 to D.K. and EYO27411 to F.S. and D.K.), the McDonnell Center for Systems Neuroscience at Washington University (to J.C-H and D.K.), the Hope Center for Neurological Disorders at Washington University (to N.S. and D.K.), and the Grace Nelson Lacy Research fund (to D.K.).

## Author contributions

J-C.H. and D.K. conceived the project. J.C.-H. performed and analyzed the functional two-photon imaging experiments. N.S. performed intra-vitreal injections and analyzed anatomical experiments. F.S. generated adeno-associated viruses. J.C.-H and D.K. wrote the manuscript with input from all the authors.

## Competing interests

The authors declare no competing interests.
