## [Peer Review File · Nature Communications]

Distributed feature representations of natural stimuli across parallel retinal pathwaysREVIEWER COMMENTS

Reviewer #1 (Remarks to the Author):

In this study, Hsiang and colleagues measured the light response properties of On-type bipolar cells in the mouse retina. They show a previously unappreciated diversity in spatiotemporal tuning properties of these bipolar cells. Further, the study strongly contributes to our understanding of bipolar cell computation by characterizing the diversity in On bipolar cell signaling with amacrine cell input intact.

Major concerns:

Measurement of the temporal tuning functions is difficult due to the limitations of Ca^{++} indicators, which have poor kinetic properties. The conclusions made in the paper about the temporal tuning seem a bit strong given these experimental limitations. Discussing the caveats of this experiment in the text would help orient the reader to potential over interpretation of the results. Also, more careful quantification of these results would further aid the reader (see below).

Minor concerns:

1. The authors rely heavily on MDS analysis, but this is not well explained or justified. Why is this approach used vs a mathematically tractable approach such as SVD/PCA. A brief summary of this technique and the pros/cons relative to other methods would be helpful for many readers.
2. The observations about the temporal frequency tuning of the different bipolar cell types is mostly qualitative in the current reading. The argument that some types are low-pass filtering while others are bandpass filtering would be strengthened if the authors fit a bandpass tuning function to their data and reported the weights for the lowpass and highpass coefficients and the cutoff frequencies for each of those components.
3. Lines 239-243. (The following comment is not a request for new analysis, but purely a suggestion that the authors can do with as they like.) The findings reported here suggest that spatial integration in the On bipolar cell RF center is relatively linear. One could interpret this result to suggest that the spatiotemporal receptive fields of the bipolar cells should be fairly effective in predicting the cells' responses to the natural movies (from the LN model). If this were not the case, then it would suggest that some combination of nonlinear integration from the surround and adaptation within the bipolar

cells are not captured by the LN model, but are engaged for natural stimuli. This could be tested with the data already collected. Is the LN model more accurate in some bipolar types than others at predicting responses to natural movies? This would suggest further diversity in computation across types.

Reviewer #2 (Remarks to the Author):

Hsiang et al. conducted Ca imaging from all types of ON bipolar cells in the mouse retina. The authors generated transgenic mice that express GCamp6f by ON bipolar cells and observed light-evoked responses in their axon terminals. They distinguished four types of type-5 bipolar cells and type-6, 7, and 8/9 cells, which are consistent with previous papers. Then, they used artificial patterns to examine temporal and spatial features and found that each type of bipolar cell exhibits unique responses. Furthermore, they tested naturalistic stimuli and ON bipolar cell responses. They found that spatial contrast and temporal contrast differentiate type-specific responses. Finally, the authors tested whether axon terminals of single bipolar cells respond to stimuli homogeneously or segment-dependently. They found that axons' responses were homogeneous across arbors, indicating that visual processing occurs in single bipolar cells as a unit.

This article contains detailed physiological and morphological characteristics of bipolar cell types, highly significant and valuable knowledge addition to the field. It is overall great work with great details. Some strengths and concerns are described below.

Strengths

- Bipolar cell types are well-defined with concrete strategies. Type 5o and 5i, and type 7 and 8/9 are similar to each other; however, the authors utilized CAR staining and measuring spatial extent to distinguish them elegantly. Based on the Figure 1 images, types 7 and 8/9 might be able to be distinguished by the IPL depths. The main axon terminals of type 7 ramify closer to the ON Chat band than type 8/9 terminals.
- Multidimensional scaling (MDS) was conducted to differentiate functional and morphological datasets from bipolar cells. The distance among 7 bipolar cell types shows that the clustering is convincing.
- The natural scene movie clips were recorded from a mouse's visual perspective. The angle of the mouse fixation position was matched along with the locomotion speeds.

Weaknesses

- The temporal precision of GCamp6f response is a concern. The authors show a simulation of GCamp6f signals in response to a high-frequency stimulus (Suppl Fig 2), which indicates that the GCamp signal

drastically decreases above 4 Hz stimulus. However, temporal features of bipolar cells are compared at 4 and 8 Hz. Even though the corrections are made, small signals might not be accurate. That point is not fully addressed.

- Furthermore, some bipolar cells are reported to exhibit regenerative spikes (Pan and Hu, 2000; Baden et al., 2013; Hellmer et al., 2016). However, except for the XBC, fast temporal features were not observed in Figure 2. Please address any spike activities in some ON bipolar cells. It is a concern if GCamp6f responds fast enough to exhibit the regenerative event.
- Figure 6 shows feature encoding among ON bipolar cells. To compare feature sensitivity across ON bipolar cell types, having a panel of all types in this figure would be great.
- Figures 7 and 8 show an amazing result of homogenous processing in individual bipolar cell axons. The result addresses recent reports of heterogenous axon terminal responses to visual stimuli. Another thought is that different bipolar cells in the same region respond differently. Figure 5 results indicate that this is the case. However, each type exhibits a range of response. Please address this concern.
- Data in Figure 8 suggest a large receptive field center exists in some types of ON bipolar cells, such as type 5t, 6-8/9. The dendritic field size is reported to be within 100 μm , implying that there must be some mechanisms to enlarge the receptive field sizes. Please address this issue.

Baden T, Berens P, Bethge M, Euler T (2013) Spikes in mammalian bipolar cells support temporal layering of the inner retina. *Curr Biol* 23:48-52.

Hellmer CB, Zhou Y, Fyk-Kolodziej B, Hu Z, Ichinose T (2016) Morphological and physiological analysis of type-5 and other bipolar cells in the Mouse Retina. *Neuroscience* 315:246-258.

Pan ZH, Hu HJ (2000) Voltage-dependent Na(+) currents in mammalian retinal cone bipolar cells. *J Neurophysiol* 84:2564-2571.

Reviewer #3 (Remarks to the Author):

Review of NCOMMS-23-37106

Distributed feature representations of natural stimuli across parallel retinal pathways

This paper describes a range of experiments focused on measuring calcium signals in the axonal arbours of bipolar cells in the mouse retina. The authors try to make general statements about patterns of functional organization from a collection of variable responses, but this is not convincing as revealing general principles. The conclusion in the abstract of “upper block: diverse temporal, uniform spatial tuning; lower block: diverse spatial, uniform temporal tuning” seems to be based on one or two apparent outliers. The overriding picture is of difficulties in discerning general patterns in the functional organization of bipolar cells and then making of artificial distinctions. Other aspects of the paper, such as comparison of responses to “artificial” and “naturalistic” stimuli do not provide any new insights into retinal function.

Major points

Temporal frequency tuning across ON bipolar pathways

I didn't find this section at all convincing, as described below. The presentation is imprecise and vague. It is difficult to see how the main conclusion is justified (“Both outliers are localized in the upper block, generating diversity in temporal processing within this block, whereas temporal processing is uniform in the lower block”).

In Fig. 2a, a dominant feature of the response in all BC types except XBC is that the baseline GCaMP signal shifts downwards with sinusoidal stimulation. This is never commented on but should be because it is an essential feature of the response that will affect synaptic transmission from ON bipolar cells and what we can learn about axons containing different computational units.

The suppression by contrast also indicates that the 150 μm spot stimulus is not simply activating the receptive field “center:” it is also causing relatively strong hyperpolarization of bipolar cell axons, presumably by activating inhibitory inputs from amacrine cells. The text implies that the inhibitory surround is only activated significantly by the larger (800 μm) spot, but this is not the case and should be made clear.

A comparison of responses to the 150 μm and 800 μm spots is fundamental to interpreting Fig. 2c and also relates to basic conclusions of the paper that build from this figure, but the responses to the 800 μm spots are not shown in the main paper. I think that responses to spots of different sizes should be overlaid in Fig. 2a to allow the reader to judge quite how spot size affects responses of different types of bipolar cell.

The authors are clearly aware of the difficulties in determining frequency tuning by measurement of GCaMP signals. They try to correct for the impact of the relatively slow frequency response of the reporter using a correction procedure shown in Supplementary Figure 2 and described in Methods. This procedure assumes that the decay time-constant of the CGaMP signal is fixed, but in fact it will depend on the volume of the axonal compartment being analyzed (reference 68). It is striking that the XBC type of bipolar cell has the smallest boutons on the axonal arborization (Fig. 1b) as well as the ability to follow higher temporal frequencies (Fig. 1c). This is exactly the correlation expected from the work of Baden et al, simply reflecting variations in bouton size. It is essential that the effects of compartment volume be taken into account when comparing the dynamics of the GCaMP signal in different types of bipolar cell.

I don't understand the legend to Fig. 2c. The word "variance" has a specific statistical meaning: how does that relate to measurements? I could not find this information in the main text.

Line 108. "Thus, the temporal frequency response functions of six of the eight parallel ON bipolar pathway overlap with two opposing outliers (BC5t and XBC)". I found this hard to understand. First, I count 7 types of bipolar cell in Figures 1 and 2 and only 7 are listed in line 77. Second, what does "overlap with two opposing outliers" mean? The language is vague.

The main conclusion of this section is that "Both outliers are localized in the upper block, generating diversity in temporal processing within this block, whereas temporal processing is uniform in the lower block". I could see only one of the 7 types of bipolar cell that obviously generates axonal responses different to the others – the XBC. The B5Ct looks qualitatively similar to the others in Fig. 2a, b and c. Further, there is no attempt to provide some sort of statistical justification for categorizing the XBC and B5Ct as being different to the other five.

Divergence of temporal and spatial information in ON bipolar pathways

Line 129. "Thus, an unbiased analysis of the encoding space of ON bipolar cells revealed that differences in their responses to an artificial stimulus are dominated by differences in spatial and temporal tuning". This sentence is a (trivial) tautology. I do not understand what has been learnt from Fig. 4 and this section of the paper.

Distributed encoding of naturalistic stimulus/Homogeneous processing of naturalistic stimuli in individual bipolar cell axons

The main conclusion from these sections is stated in line 16 of the abstract as “ON bipolar cells extract temporal and spatial features similarly from artificial and naturalistic stimuli.” But why would they not? Unless they rewire when the retina finds out what the experimenter is delivering to it. Again, I find it hard to see any significant advance in understanding.

Homogenous processing of artificial stimuli in individual bipolar cell axons.

Scholarship

The paper is written in a “mouse-centric” manner that artificially ignores large bodies of earlier work on retinal function. This approach, emphasizing the species over the biological process, undermines scholarship.

Line 50. “It has been suggested that amacrine cell influences diversify the responses within a single bipolar cell axon²³”. The generation of orientation selectivity in bipolar cell axon terminals was first demonstrated in zebrafish

Antinucci, P., Suleyman, O., Monfries, C., and Hindges, R. (2016b). Neural mechanisms generating orientation selectivity in the retina. *Curr. Biol.* 26, 1802–1815. doi: 10.1016/j.cub.2016.05.035

Johnston J, Seibel SH, Darnet LSA, Renninger S, Orger M, Lagnado L. A Retinal Circuit Generating a Dynamic Predictive Code for Oriented Features. *Neuron*. 2019 Jun 19;102(6):1211-1222.e3. doi: 10.1016/j.neuron.2019.04.002.

Line 115. In mammalian retinas, bipolar cell axons with transient responses are thought to stratify toward the center of the IPL, and bipolar cell axons with sustained responses toward the IPL borders 6,11,37–41. This idea was established in earlier in non-mammalian retinas (for instance the work of Sam Wu using salamander retina). A good review that summarizes earlier work on the organization of temporal processing in the retina is: Baccus SA. Timing and computation in inner retinal circuitry. *Annual Review of Physiology*. 69: 271-90. PMID 17059359 DOI: 10.1146/annurev.physiol.69.120205.124451

Minor points

Abstract

Line 13. "We discover that retinal ON bipolar cells (second-order neurons of the visual system) are divided into two blocks of four types that distribute temporal and spatial information encoding, respectively." Needs to be worded more clearly: the "respectively" pertains to the two blocks, not the four types.

Line 16. "ON bipolar cells extract temporal and spatial features similarly from artificial and naturalistic stimuli." But why would they not? Unless they rewire when the retina finds out what the experimenter is delivering to it.

Line 49. "Bipolar cell axons integrate photoreceptor signals from their dendrites in the outer retina with local inputs from amacrine cells." The axons do not do this.

Line 54. What is meant by "whether axonal organization varies or is shared across bipolar cell types"? Different types of BC have different axonal arborizations so their organization is not shared. Is something more subtle meant here?

Line 89. "ON bipolar cells are thought to vary in their preferences for temporal stimulus contrast 11,21." Variations in the frequency tuning of bipolar cells has been observed over several decades in a range of species. The word "thought"

Line 91. "150- μ m". There should be a simple space between number and units. Dash is non-standard and unhelpful.

Line 119. "First, the responses of axons in the upper block were, on average, more transient than those in the lower block (Fig. 3a-c, $p < 0.001$ by one-sided Wilcoxon rank sum, upper block: $n = 27$, lower block: $n = 20$, $r = 0.61$, $z = 4.19$)." Which size spots are being compared? What is the metric of transience?

Reviewer #4 (Remarks to the Author):

This manuscript reports on the response properties of ON-type bipolar cells in the mouse retina. First, the authors use sparse ON BC labeling and demonstrate that all ON BC types can be distinguished in their experiments based on anatomical features. They then use dense ON BC GCaMP6f expression to measure calcium responses in axonal arbors and compare response properties between apparent types at the population level. These analyses are well designed and type-specificity of the findings is credible.

Based on the measured responses, the authors functionally organize the complement of ON BC types into two blocks, distal and proximal, anatomically separated by the ON starburst amacrine cell dendritic layer. They argue that selective visual processing is distributed across these two blocks (distal: spatially uniform, temporally diverse; proximal: spatially diverse, temporally uniform), and this is reasonably supported by the data. To test generality of this claim and avoid bias from the use of artificial visual stimuli, the authors generated 'mouse-cam'-style videos and used those to stimulate the retina while recording ON bipolar cell population responses. Responses are analyzed with appropriate means including analysis of response correlations with feature representation-type transformations of the videos. The results of this analysis support the main claims. Finally, comparison of calcium responses within individual axonal branches of single cells showed uniform signaling across the axonal arbor. The noted absence of evidence for subcellular, localized computation is relevant following recent claims to the contrary in directional motion encoding.

A temporal division of signaling with transient responses near the central IPL and sustained responses proximal was already known but spatial tuning and cell-type specific detail was lacking. The comprehensive mapping done here resolves all of that, and also shows one exception to the temporal rule (the BC5t type is sustained within the presumed transient zone), and shows how spatial response properties co-organize with it. The study gives useful and apparent complete information about ON BC organization across IPL layers, and this is in my opinion is the main contribution of the work.

Comments.

1. During stimulation with naturalistic movies, receptive field position within the imaged area mattered: because the movie was not spatially uniform, different parts of the image contain different information/features, and this would create differences in the responses of simultaneously recorded BCs at different spatial positions within the imaged field. How large was the imaged area? How did you account for the fact that not all BCs received the same visual stimulation? This must be addressed, for example in Results, around Line 195.

2. Line 245-246 is cryptic and should be elaborated: what does it mean that 'motion provides an independent feature [...] that is distributed across ON BCs in both blocks'?

3. Line 294-295, the claim that 'for naturalistic movies, ON bipolar cells vary systematically in their responses to coherent motion' needs to be better substantiated in Results, with references to the data/figures.

Minor.

4. For 'split into ON and OFF channels' (line 35-36) consider including reference to Ratliff, ..., Balasubramanian, 2010, which demonstrates the benefit of the specific (uneven) ON-OFF split within the mammalian retina with regards to natural scene encoding.

5. Fig. 1f legend text typo: '...profiles OF all ON bipolar... '.

6. Fig. 2B: colors of the traces representing different BC types are difficult to discern. Consider broader color scale and/or adding symbols.

7. Fig. 3a legend text, typo: '... to stimuli spots...' correct.

Reviewer #1 (Remarks to the Author):

In this study, Hsiang and colleagues measured the light response properties of On-type bipolar cells in the mouse retina. They show a previously unappreciated diversity in spatiotemporal tuning properties of these bipolar cells. Further, the study strongly contributes to our understanding of bipolar cell computation by characterizing the diversity in On bipolar cell signaling with amacrine cell input intact.

We thank the reviewer for carefully reading our manuscript and for their positive remarks. As detailed in our point-by-point responses, we have performed additional analyses and rewritten sections of the manuscript following the reviewer's suggestions. Our new analyses confirm and strengthen our previous conclusions. We are grateful to the reviewer for helping us improve our manuscript.

Major concerns:

Measurement of the temporal tuning functions is difficult due to the limitations of Ca^{++} indicators, which have poor kinetic properties. The conclusions made in the paper about the temporal tuning seem a bit strong given these experimental limitations. Discussing the caveats of this experiment in the text would help orient the reader to potential over interpretation of the results. Also, more careful quantification of these results would further aid the reader (see below).

We thank the reviewer for raising this important point. In our initial submission, we modeled the impact of the GCaMP6f indicator kinetics on the observed signals (Supplementary Fig. 2) but did not explicitly test the reliability with which the observed signals report stimulus responses. In our revisions, we generated GCaMP6f traces based on the autocorrelation function of each region of interest (ROI). We then compared the repeat reliabilities of these simulated signals to those of the observed signals. Up to and including 8 Hz stimuli, the observed responses of more than half of the ROIs were above the 95th percentile of the repeat reliabilities of the simulated signals. This makes us confident in our conclusions about differences in the cell-type-specific tuning functions of ON bipolar cells between 0.5 and 8 Hz. Notably, these tuning functions diverge mostly between 2 and 4 Hz (i.e., in a range with high confidence). We have added a new Supplementary Figure (Supplementary Fig. 3) summarizing the comparisons between simulated and observed repeat reliabilities at different stimulus frequencies.

In addition to the statistical comparisons in our initial submission showing that XBCs and BC5t differ from the rest of the ON bipolar cells as a group, we include comparisons among all individual ON bipolar cell types (by bootstrapping) in our revised manuscript. This confirms that the temporal tuning functions of XBC differ significantly from all other types, while the tuning functions of BC5t differ significantly from BC5i, XBC, and BC8/9. We include this analysis in our new Supplementary Figure (Supplementary Fig. 3) in our revised manuscript.

Minor concerns:

1. The authors rely heavily on MDS analysis, but this is not well explained or justified. Why is this approach used vs a mathematically tractable approach such as SVD/PCA. A brief summary of this technique and the pros/cons relative to other methods would be helpful for many readers.

As suggested by the reviewer, we have expanded our explanation of our MDS analyses and added a brief justification for choosing MDS over other approaches (e.g., PCA) in the Methods

section of our revised manuscript. There are two main reasons for choosing MDS over PCA (or similar approaches). MDS analyses start by computing a distance matrix (i.e., a measurement summarizing the dissimilarity between cells based on their complete response traces). It then identifies the dimensions of the space defined by the distance matrix across which the cell population varies the most (i.e., the principal coordinates). Thus, our analysis focuses on what organizes the diversity of ON bipolar cells (i.e., the goal of our study). PCA (and similar approaches) instead highlights the stimulus dimensions across which responses vary the most. Thus, its primary focus is on the stimulus space rather than the cellular encoding space.

In addition to this conceptual difference, there is a technical reason we chose MDS over PCA to analyze responses to naturalistic stimuli. We presented naturalistic stimuli as two to four repeats of 11 representative movie clips. To include a cell's response to a given clip in our analysis, it needed to exceed a quality threshold (i.e., repeat reliability). Because MDS starts by calculating distances between cell pairs, each distance calculation could include all clips for which both cells exceeded the required repeat reliability. Different clips could be included for different cell pairs. In PCA, all cells need to be compared across the same stimulus clips, which would either compromise response quality or drastically reduce the number of admissible clips.

2. The observations about the temporal frequency tuning of the different bipolar cell types is mostly qualitative in the current reading. The argument that some types are low-pass filtering while others are bandpass filtering would be strengthened if the authors fit a bandpass tuning function to their data and reported the weights for the lowpass and highpass coefficients and the cutoff frequencies for each of those components.

We agree with the reviewer that our previous distinction between band-pass and low-pass filtering was purely qualitative. In our revised manuscript, we abandon this distinction, because it only applies to a specific frequency range (0.5 - 8 Hz) and thus might be misleading. If we extended stimuli into a lower frequency range, all ON bipolar cells would likely act as band-pass filters.

To quantitatively support our assertion that XBC and BC5t are outliers in their temporal tuning, we (1) retain our previous comparison of their responses to stimuli from 2 - 8 Hz to the other ON bipolar cells as a group by Wilcoxon signed-rank tests. In our revised manuscript, we (2) added pairwise comparisons among all ON bipolar cell types by bootstrapping. This confirmed that XBC differs significantly from all other types, and BC5t differs significantly from BC5i, XBC, and BC8/9. The remaining ON bipolar cell types do not differ significantly in their temporal tuning. We present this quantification in a new Supplementary Figure (Supplementary Fig. 3) in our revised manuscript.

3. Lines 239-243. (The following comment is not a request for new analysis, but purely a suggestion that the authors can do with as they like.) The findings reported here suggest that spatial integration in the On bipolar cell RF center is relatively linear. One could interpret this result to suggest that the spatiotemporal receptive fields of the bipolar cells should be fairly effective in predicting the cells' responses to the natural movies (from the LN model). If this were not the case, then it would suggest that some combination of nonlinear integration from the surround and adaptation within the bipolar cells are not captured by the LN model, but are engaged for natural stimuli. This could be tested with the data already collected. Is the LN model more accurate in some bipolar types than others at predicting responses to natural movies? This would suggest further diversity in computation across types.

We thank the reviewer for their suggestion. We agree that because we show that ON bipolar cells integrate spatial information relatively linearly, an LN model should be able to predict their responses reasonably well. Indeed, we (and others before us) find this to be true for artificial white noise stimuli. There are significant challenges in estimating the parameters of an LN model from naturalistic stimuli. LN models constructed from ON bipolar cell responses to white noise stimuli often do well at predicting responses to naturalistic movies but are somewhat inconsistent. We are exploring the reasons for this inconsistency but think that this is beyond the scope of the present study. We chose not to include LN model analyses in our manuscript because we wanted to focus on our innovative model-free analyses of the encoding differences between bipolar cells.

Reviewer #2 (Remarks to the Author):

Hsiang et al. conducted Ca imaging from all types of ON bipolar cells in the mouse retina. The authors generated transgenic mice that express GCamp6f by ON bipolar cells and observed light-evoked responses in their axon terminals. They distinguished four types of type-5 bipolar cells and type-6, 7, and 8/9 cells, which are consistent with previous papers. Then, they used artificial patterns to examine temporal and spatial features and found that each type of bipolar cell exhibits unique responses. Furthermore, they tested naturalistic stimuli and ON bipolar cell responses. They found that spatial contrast and temporal contrast differentiate type-specific responses. Finally, the authors tested whether axon terminals of single bipolar cells respond to stimuli homogeneously or segment-dependently. They found that axons' responses were homogeneous across arbors, indicating that visual processing occurs in single bipolar cells as a unit.

This article contains detailed physiological and morphological characteristics of bipolar cell types, highly significant and valuable knowledge addition to the field. It is overall great work with great details. Some strengths and concerns are described below.

Strengths

- Bipolar cell types are well-defined with concrete strategies. Type 5o and 5i, and type 7 and 8/9 are similar to each other; however, the authors utilized CAR staining and measuring spatial extent to distinguish them elegantly. Based on the Figure 1 images, types 7 and 8/9 might be able to be distinguished by the IPL depths. The main axon terminals of type 7 ramify closer to the ON Chat band than type 8/9 terminals.
- Multidimensional scaling (MDS) was conducted to differentiate functional and morphological datasets from bipolar cells. The distance among 7 bipolar cell types shows that the clustering is convincing.
- The natural scene movie clips were recorded from a mouse's visual perspective. The angle of the mouse fixation position was matched along with the locomotion speeds.

We thank the reviewer for their careful reading and insightful summary of the conclusions and strengths of our manuscript. As detailed in our point-by-point responses, we have performed new analyses and rewritten sections of our manuscript to address weaknesses identified by the reviewer in our point-by-point responses below. Our new analyses confirm and strengthen our previous conclusions. We are grateful to the reviewer for helping us improve our manuscript.

Weaknesses

- The temporal precision of GCamp6f response is a concern. The authors show a simulation of GCamp6f signals in response to a high-frequency stimulus (Suppl Fig 2), which indicates that the GCamp signal drastically decreases above 4 Hz stimulus. However, temporal features of bipolar cells are compared at 4 and 8 Hz. Even though the corrections are made, small signals might not be accurate. That point is not fully addressed

We thank the reviewer for raising this critical point. In our initial submission, we examined the influence of the GCaMP6f indicator kinetics on the observed signals (Supplementary Fig. 2). However, we did not explicitly test the reliability with which the observed signals report stimulus responses. In our revisions, we generated GCaMP6f traces based on the autocorrelation function of each region of interest (ROI). Subsequently, we compared the repeat reliabilities of these simulated signals to those of the observed signals. Up to and including 8 Hz stimuli, the observed responses of more than half of the ROIs surpassed the 95th percentile of the repeat reliabilities of the simulated signals. These observations strongly support our conclusions about differences

in the cell-type-specific tuning functions of ON bipolar cells between 0.5 and 8 Hz. Notably, these tuning functions diverge mostly between 2 and 4 Hz, where our confidence is particularly high. We include a new Supplementary Figure (Supplementary Fig. 3) that contrasts the repeated reliability of simulated and observed signals across varied stimulus frequencies.

In addition to the statistical comparisons in our initial submission showing that XBCs and BC5t differ from the rest of the ON bipolar cells as a group, we include comparisons among all individual ON bipolar cell types (by bootstrapping) in our revised manuscript. This confirms that the temporal tuning functions of XBC differ significantly from all other types, while the tuning functions of BC5t differ significantly from BC5i, XBC, and BC8/9. We include this analysis in our new Supplementary Figure (Supplementary Fig. 3) in our revised manuscript.

- Furthermore, some bipolar cells are reported to exhibit regenerative spikes (Pan and Hu, 2000; Baden et al., 2013; Hellmer et al., 2016). However, except for the XBC, fast temporal features were not observed in Figure 2. Please address any spike activities in some ON bipolar cells. It is a concern if GCamp6f responds fast enough to exhibit the regenerative event.

Following the reviewer's recommendation, we carefully searched for evidence of spiking activities in our ON bipolar cell recordings:

(1) We analyzed the response amplitude distributions for sequential stimulus presentations (square-wave-modulated spots of varying size). If ON bipolar cells spike, one would expect response amplitudes to have binomial distributions composed of discrete events, non-events, and maybe multiples of events. By contrast, if ON bipolar cells do not spike, one would expect response amplitudes to have continuous distributions. We find that all ON bipolar cell types have continuous response amplitude distributions to sequential stimulus presentations.

(2) We searched for spontaneous spike events during the stimulus OFF phase of square-wave-modulated spots of varying size (i.e., events with an amplitude >80% of the response amplitude during the stimulus ON phase). We found no such event in 3,215 trials across 49 bipolar cells.

Together, these findings suggest that ON bipolar cells did not spike in our recordings. We present our new analyses in the Results section and a new Supplementary Figure (Supplementary Fig. 7) in our revised manuscript.

- Figure 6 shows feature encoding among ON bipolar cells. To compare feature sensitivity across ON bipolar cell types, having a panel of all types in this figure would be great.

We agree with the reviewer that it is important to compare feature sensitivities across ON bipolar cell types. Given space limitations in Figure 6, we compare sensitivities to luminance in Figure 5 and sensitivities to spatial contrast, temporal contrast, and coherent motion in Supplementary Fig. 14 in our revised manuscript.

- Figures 7 and 8 show an amazing result of homogenous processing in individual bipolar cell axons. The result addresses recent reports of heterogenous axon terminal responses to visual stimuli. Another thought is that different bipolar cells in the same region respond differently. Figure 5 results indicate that this is the case. However, each type exhibits a range of responses. Please address this concern.

We thank the reviewer for highlighting the significance of our observation that visual processing within ON bipolar cell axon arbors is homogeneous. As the reviewer correctly points out, there is some variability in the responses of ON bipolar cells of the same type. Notably, this variability is smaller than the functional differences between ON bipolar cell types. To what extent the source of the variability in responses of same-type ON bipolar cells is biological vs. experimental is difficult to discern. In one instance, we recorded the activity of two bipolar cells of the same type in the same image. The responses of these two cells (Fig. R1) were more similar than the average same-type pairs recorded in different images, suggesting that at least some of the variability is likely experimental, consistent with previous observations (e.g., Kalmar et al., *Journal of Neuroscience*, 2002; Zhao et al., *Scientific Reports*, 2020; Soto et al., *Neuron*, 2020). Because our insights into the source of variability are limited, we reference the studies above and mention the likely partially experimental origin in the Result section of our revised manuscript.

- Data in Figure 8 suggest a large receptive field center exists in some types of ON bipolar cells, such as type 5t, 6-8/9. The dendritic field size is reported to be within 100 μm , implying that there must be some mechanisms to enlarge the receptive field sizes. Please address this issue.

Figures 3 and 8 show the responses of the different ON bipolar cell types to square-wave-modulated (1.5 s ON, 1.5 s OFF) spots of increasing size. In all cases, responses peak at $\leq 100 \mu\text{m}$ spot size. For several cell types, including BC5t, BC6, and BC8/9, larger spots still elicit strong responses. This does not reflect an extension of the receptive field center beyond the dendritic field, but rather the weak surround inhibition of these cells. The spot size eliciting peak responses in our study closely matches previous observations (Schwartz et al., *Nature Neuroscience*, 2012; Borghuis et al., *Journal of Neuroscience*, 2013; Purgert et al., *Journal of Neurophysiology*, 2015; Franke et al., *Nature*, 2017).

Reviewer #3 (Remarks to the Author):

This paper describes a range of experiments focused on measuring calcium signals in the axonal arbours of bipolar cells in the mouse retina. The authors try to make general statements about patterns of functional organization from a collection of variable responses, but this is not convincing as revealing general principles. The conclusion in the abstract of “upper block: diverse temporal, uniform spatial tuning; lower block: diverse spatial, uniform temporal tuning” seems to be based on one or two apparent outliers. The overriding picture is of difficulties in discerning general patterns in the functional organization of bipolar cells and then making of artificial distinctions. Other aspects of the paper, such as comparison of responses to “artificial” and “naturalistic” stimuli do not provide any new insights into retinal function.

We thank the reviewer for their careful reading and constructive criticism of our manuscript. As detailed in our point-by-point response below, we have performed additional analyses, made changes to our main figures, added new supplementary figures, and rewritten sections of our manuscript to address the reviewer’s concerns. Our new analyses strengthen and extend our previous findings and support our overarching conclusion that the ON bipolar pathway in mice is divided into two blocks that distribute encoding of temporal and spatial information, respectively. We are grateful to the reviewer for helping us improve our manuscript.

Major points

Temporal frequency tuning across ON bipolar pathways

I didn’t find this section at all convincing, as described below. The presentation is imprecise and vague. It is difficult to see how the main conclusion is justified (“Both outliers are localized in the upper block, generating diversity in temporal processing within this block, whereas temporal processing is uniform in the lower block”).

To provide additional support for our conclusions, we added the following analyses:

(1) We confirmed that only BC5t and XBC differ significantly in their temporal tuning to sinusoidal flicker stimuli from all other ON bipolar cells as a group (by Wilcoxon signed-rank test). We present the statistics for this in the Results section of our revised manuscript. We also compared the tuning functions of all ON bipolar cell types pairwise (by bootstrapping), demonstrating that XBC differs significantly from all other types and BC5t differs significantly from BC5i, XBC, and BC8/9. No other bipolar cell types differ significantly from each other. We include this analysis in our new Supplementary Figure (Supplementary Fig. 3) in our revised manuscript.

(2) We compared the differences in the surround strength and response transience of different-type cell pairs in the upper block vs. the lower block. This analysis showed that surround strength varied significantly more between cell pairs in the lower than the upper block (Kolmogorov-Smirnov test, $p < 0.001$, $d = 0.32$), whereas response transience varied significantly more in the upper than the lower block (Kolmogorov-Smirnov test, $p < 0.001$, $d = 0.28$). We have added this analysis to Figure 3 (panel d) and the Results section of our revised manuscript.

In Fig. 2a, a dominant feature of the response in all BC types except XBC is that the baseline GCaMP signal shifts downwards with sinusoidal stimulation. This is never commented on but should be because it is an essential feature of the response that will affect synaptic transmission from ON bipolar cells and what we can learn about axons containing different computational units.

We agree that the baseline shift in response to flicker stimuli is an interesting observation in our study. Suppressive responses are unique to the flicker stimulus and may result from its specific (and unnatural) temporal structure. Following the reviewer's recommendation, we highlight this observation in the Results section of our revised manuscript. We will explore the circuit mechanisms underlying the baseline shift to flicker stimuli in a future study but believe that this is beyond the scope and outside the focus of our present study on the distributed encoding of naturalistic stimuli across the ON bipolar cell population.

The suppression by contrast also indicates that the 150 μm spot stimulus is not simply activating the receptive field "center:" it is also causing relatively strong hyperpolarization of bipolar cell axons, presumably by activating inhibitory inputs from amacrine cells. The text implies that the inhibitory surround is only activated significantly by the larger (800 μm) spot, but this is not the case and should be made clear.

We agree with the reviewer about the organization of the bipolar cell receptive field, which can be approximated as the sum of two Gaussians: a narrow excitatory (or activating) one and a broader inhibitory (or suppressive) one. The centers of both Gaussians are aligned. Some authors (incl. the reviewer) use the term surround to refer to the broad inhibitory Gaussian. In this usage, small stimuli centered on the cell activate the surround. Indeed, they activate the surround more than stimuli that are displaced laterally. By contrast, other authors (including us) refer to the surround as the area of the receptive field in which the broader inhibitory Gaussian outweighs the narrow excitatory Gaussian, such that stimuli in this region suppress the response (see also Kuffler, *Journal of Neurophysiology*, 1953). In this usage, small stimuli centered on the cell do not engage the surround.

Prompted by the reviewer's comment, we have clarified our usage of the term surround in the Results section of our revised manuscript.

A comparison of responses to the 150 μm and 800 μm spots is fundamental to interpreting Fig. 2c and also relates to basic conclusions of the paper that build from this figure, but the responses to the 800 μm spots are not shown in the main paper. I think that responses to spots of different sizes should be overlaid in Fig. 2a to allow the reader to judge quite how spot size affects responses of different types of bipolar cell.

Following the reviewer's suggestions, we overlaid responses to 150 μm and 800 μm spots; we found the resulting figure too crowded and difficult to parse visually. Therefore, we show response traces to 800 μm stimuli in a Supplementary Figure (Supplementary Fig. 6). However, we show summary data comparing the temporal tuning of ON bipolar cells to large and small stimuli in Figure 2 (panel c). We hope that the reviewer is okay with this choice.

The authors are clearly aware of the difficulties in determining frequency tuning by measurement of GCaMP signals. They try to correct for the impact of the relatively slow frequency response of the reporter using a correction procedure shown in Supplementary Figure 2 and described in Methods. This procedure assumes that the decay time-constant of the CGaMP signal is fixed, but in fact it will depend on the volume of the axonal compartment being analyzed (reference 68). It is striking that the XBC type of bipolar cell has the smallest boutons on the axonal arborization (Fig. 1b) as well as the ability to follow higher temporal frequencies (Fig. 1c). This is exactly the correlation expected from the work of Baden et al, simply reflecting variations in bouton size. It is

essential that the effects of compartment volume be taken into account when comparing the dynamics of the GCaMP signal in different types of bipolar cell.

We thank the reviewer for raising this important point. We performed several new analyses to further characterize the temporal tuning of ON bipolar cell axon responses, test their relationships to anatomical features, and confirm the robustness of our findings to indicator kinetics.

(1) We measured the diameters of ROIs from our morphological segmentations of ON bipolar cell axons. This revealed that the ROIs of XBC, which prefer the fastest stimulus frequencies of all ON bipolar cells, did not differ significantly from the ROI size of the other ON bipolar cells. In contrast, the ROIs of BC5t, which prefer the slowest stimulus frequencies of all ON bipolar cells, were smaller than the ROIs of the other ON bipolar cells, contrary to expectations based on changes in the surface/volume ratio. When we split the distribution of all ROIs into a large and a small half (i.e., separated at the median ROI diameter), we found that their temporal tuning was indistinguishable. We show these analyses in a new Supplementary Figure (Supplementary Fig. 4) in our revised manuscript.

(2) We also divided the ROIs of each ON bipolar cell type into a large and small half and compared their temporal tuning. The temporal tuning of large and small ROIs did not differ significantly for this within-type comparison. We present these analyses in a new Supplementary Figure (Supplementary Fig. 5). Together with (1), these findings indicate that morphological features do not account for the cell-type-specific differences in the temporal frequency-response functions of mouse ON bipolar cell axons. In addition to the new Supplementary Figures, we present our findings in a new paragraph of the Results section in our revised manuscript.

(3) To address concerns about how reliably the observed GCaMP6f signals report stimulus responses given indicator kinetics, we created GCaMP6f traces based on the autocorrelation function of each ROI. We then compared the repeat reliabilities of these simulated signals to those of the observed signals. Up to and including 8 Hz stimuli, the observed responses of more than half of the ROIs surpassed the 95th percentile of the repeat reliabilities of the simulated signals. These observations strongly support our conclusions about differences in the cell-type-specific frequency tuning functions of ON bipolar cells between 0.5 and 8 Hz. Notably, these tuning functions diverge mostly between 2 and 4 Hz, where our confidence is particularly high. We include a new Supplementary Figure (Supplementary Fig. 3) comparing the repeated reliability of simulated and observed signals across varied stimulus frequencies.

I don't understand the legend to Fig. 2c. The word "variance" has a specific statistical meaning: how does that relate to measurements? I could not find this information in the main text.

We thank the reviewer for spotting this mistake. We have corrected it in the legend of Fig. 2 in our revised manuscript. *"Plots of the difference in the F1 power of responses to 150 μm (C: center, in diameter) and 800 μm (S: surround) spots across bipolar cell types, with frequencies from 0.5 to 8 Hz."*

Line 108. "Thus, the temporal frequency response functions of six of the eight parallel ON bipolar pathway overlap with two opposing outliers (BC5t and XBC)". I found this hard to understand. First, I count 7 types of bipolar cell in Figures 1 and 2 and only 7 are listed in line 77. Second, what does "overlap with two opposing outliers" mean? The language is vague.

Transcriptomic (Shekhar et al., Cell, 2016) and connectomic surveys (Greene et al., Cell Reports, 2016; Sabbah et al., bioRxiv, 2018) indicate that there are eight ON bipolar cell types in mice (BC5o, BC5i, BC5t, XBC, BC6, BC7, BC8, and BC9). In our analyses, we combine data from BC8 and BC9 (BC8/9). That is why there are seven groups in Figures 1, 2, and 7. We clarified the classification of ON bipolar cells and our combination of BC8 and BC9 in the Results section of our revised manuscript.

Following the reviewer's suggestion, we have made the language describing the temporal frequency-response functions in the Results section of our revised manuscript more precise. *"The frequency-response functions of most ON bipolar cell types (i.e., BC5i, BC5o, BC6, BC7, and BC8/9) did not differ significantly, with two exceptions: XBC and BC5t (Fig. 2a, b). XBC preferred higher frequency stimuli than the rest of the ON bipolar cells (Fig. 2a, b, one-sided Wilcoxon rank sum test, $p < 0.001$, $r = 0.65$, $z = 4.08$, Supplementary Fig. 3), whereas BC5t preferred lower frequency stimuli than the rest (Fig. 2a, b, one-sided Wilcoxon rank sum test, $p < 0.01$, $r = -0.42$, $z = -2.59$, Supplementary Fig. 3)."*

The main conclusion of this section is that "Both outliers are localized in the upper block, generating diversity in temporal processing within this block, whereas temporal processing is uniform in the lower block". I could see only one of the 7 types of bipolar cell that obviously generates axonal responses different to the others – the XBC. The B5Ct looks qualitatively similar to the others in Fig. 2a, b and c. Further, there is no attempt to provide some sort of statistical justification for categorizing the XBC and B5Ct as being different to the other five.

We performed two sets of statistical analyses to examine differences in temporal tuning of ON bipolar cell types.

(1) We compared the temporal frequency responses (2 - 8 Hz) of each bipolar cell type to those of the others as a group by Wilcoxon signed-rank tests. This showed that XBC and BC5t, but no other type, are different from the rest of the ON bipolar cells. We specify the effect sizes and significance in the Results section of our revised manuscript (see also our response to the previous point).

(2) We added pairwise comparisons of temporal frequency-response functions among all ON bipolar cell types by bootstrapping. This showed that XBC differs significantly from all other types, and BC5t differs significantly from BC5i, XBC, and BC8/9. The remaining ON bipolar cell types do not differ significantly in their temporal tuning. We present this quantification in a new Supplementary Figure (Supplementary Fig. 3) in our revised manuscript.

Divergence of temporal and spatial information in ON bipolar pathways

Line 129. "Thus, an unbiased analysis of the encoding space of ON bipolar cells revealed that differences in their responses to an artificial stimulus are dominated by differences in spatial and temporal tuning". This sentence is a (trivial) tautology. I do not understand what has been learnt from Fig. 4 and this section of the paper.

Figure 4 introduces our use of MDS to identify stimulus features that organize the diversity of ON bipolar cell responses. In our MDS analyses, we compute functional distances between ON bipolar cells based on their complete response traces (in the case of Figure 4 to square-wave-modulated spots of varying size). Therefore, the outcome that response transience and surround strength organize the principal coordinates of the ON bipolar cell encoding space is not trivial (in

theory, any number of stimulus-dependent or -independent differences between ON bipolar cells could have shaped their functional diversity), and the statement cited by the reviewer is not a tautology. We have tried to clarify our conclusion and the reasons for using MDS analyses in the Results section of our revised manuscript.

It is worth noting that differences in the responses of ON bipolar cells to spots of varying size can be analyzed in several ways (we show a more traditional approach in Figure 3). However, a key advantage of our MDS approach is that it generalizes to analyses of ON bipolar cell responses to naturalistic stimuli. This allows us to compare features that organize the diversity of ON bipolar cell responses across stimuli that differ vastly in their complexity. We find that the same features dominate ON bipolar cell responses to simple artificial and complex naturalistic stimuli.

Distributed encoding of naturalistic stimulus/Homogeneous processing of naturalistic stimuli in individual bipolar cell axons

The main conclusion from these sections is stated in line 16 of the abstract as “ON bipolar cells extract temporal and spatial features similarly from artificial and naturalistic stimuli.” But why would they not? Unless they rewire when the retina finds out what the experimenter is delivering to it. Again, I find it hard to see any significant advance in understanding.

Naturalistic stimuli contain many features (e.g., coherent motion in a scene, in-center contrast, oriented edges, etc.) and combinations of features that are not present in artificial stimuli. Differences in the sensitivities to these features and feature combinations could have organized the shared encoding space of ON bipolar cell responses to naturalistic stimuli, without a need for rewiring. Notably, ganglion cells (i.e., the postsynaptic targets of bipolar cells) respond differently to naturalistic than artificial stimuli, evidenced by the failure of linear-nonlinear models fit to responses to artificial stimuli in predicting responses to naturalistic stimuli (e.g., Heitman et al., bioRxiv, 2016). We have tried to clarify the greater complexity of naturalistic stimuli and the interpretation of our MDS analyses of ON bipolar cell responses in the Results and Methods section of our revised manuscript.

Homogenous processing of artificial stimuli in individual bipolar cell axons.

Scholarship

The paper is written in a “mouse-centric” manner that artificially ignores large bodies of earlier work on retinal function. This approach, emphasizing the species over the biological process, undermines scholarship.

We thank the reviewer for bringing this to our attention. We have broadened the context of our study to include more evidence from other model organisms in the Introduction, Discussion, and References of our revised manuscript.

Line 50. “It has been suggested that amacrine cell influences diversify the responses within a single bipolar cell axon²³”. The generation of orientation selectivity in bipolar cell axon terminals was first demonstrated in zebrafish

Antinucci, P., Suleyman, O., Monfries, C., and Hindges, R. (2016b). Neural mechanisms generating orientation selectivity in the retina. *Curr. Biol.* 26, 1802–1815. doi: 10.1016/j.cub.2016.05.035

Johnston J, Seibel SH, Darnet LSA, Renninger S, Orger M, Lagnado L. A Retinal Circuit Generating a Dynamic Predictive Code for Oriented Features. *Neuron*. 2019 Jun 19;102(6):1211-1222.e3. doi: 10.1016/j.neuron.2019.04.002.

We thank the reviewer for encouraging us to broaden the context of our study. We have included the citations suggested by the reviewer in the Introduction section of our revised manuscript.

Line 115. In mammalian retinas, bipolar cell axons with transient responses are thought to stratify toward the center of the IPL, and bipolar cell axons with sustained responses toward the IPL borders 6,11,37–41. This idea was established in earlier in non-mammalian retinas (for instance the work of Sam Wu using salamander retina). A good review that summarizes earlier work on the organization of temporal processing in the retina is: Baccus SA. Timing and computation in inner retinal circuitry. *Annual Review of Physiology*. 69: 271-90. PMID 17059359 DOI: 10.1146/annurev.physiol.69.120205.124451

We thank the reviewer for pointing out this oversight. We have included references to the review by Stephen Baccus and the original work by Sam Wu's group in the Results section of our revised manuscript.

Minor points

Abstract

Line 13. "We discover that retinal ON bipolar cells (second-order neurons of the visual system) are divided into two blocks of four types that distribute temporal and spatial information encoding, respectively." Needs to be worded more clearly: the "respectively" pertains to the two blocks, not the four types.

Following the reviewer's advice, we have rearranged and rewritten the Abstract as follows: *"...discover that retinal ON bipolar cells (second-order neurons of the visual system) are divided into two blocks of four types. The two blocks distribute temporal and spatial information encoding, respectively. ON bipolar cell axons co-stratify within each block, but separate laminarly between them (upper block: diverse temporal, uniform spatial tuning; lower block: diverse spatial, uniform temporal tuning)."*

Line 16. "ON bipolar cells extract temporal and spatial features similarly from artificial and naturalistic stimuli." But why would they not? Unless they rewire when the retina finds out what the experimenter is delivering to it.

Naturalistic stimuli contain many features (e.g., coherent motion in a scene, in-center contrast, oriented edges, etc.) and combinations of features that are not present in artificial stimuli. Differences in the sensitivities to these features and feature combinations could have organized the shared encoding space of ON bipolar cell responses to naturalistic stimuli, without a need for rewiring. Notably, ganglion cells (i.e., the postsynaptic targets of bipolar cells) respond differently to naturalistic than artificial stimuli, evidenced by the failure of linear-nonlinear models fit to responses to artificial stimuli in predicting responses to naturalistic stimuli (e.g., Heitman et al., bioRxiv, 2016). We have tried to clarify the greater complexity of naturalistic stimuli and the interpretation of our MDS analyses of ON bipolar cell responses in the Results and Methods section of our revised manuscript.

Line 49. “Bipolar cell axons integrate photoreceptor signals from their dendrites in the outer retina with local inputs from amacrine cells..” The axons do not do this.

To clarify, we have rewritten this statement as follows: *“Bipolar cell axons integrate photoreceptor signals relayed by their dendrites from the outer retina with local input from amacrine cells.”*

Line 54. What is meant by “whether axonal organization varies or is shared across bipolar cell types”? Different types of BC have different axonal arborizations so their organization is not shared. Is something more subtle meant here?

We have removed this phrase from our revised manuscript.

Line 89. “ON bipolar cells are thought to vary in their preferences for temporal stimulus contrast 11,21.” Variations in the frequency tuning of bipolar cells has been observed over several decades in a range of species. The word “thought”

We have rewritten this as follows: *“ON bipolar cells vary in their preferences for temporal stimulus contrast.”*

Line 91. “150-μm”. There should be a simple space between number and units. Dash is non-standard and unhelpful.

We have corrected this throughout our revised manuscript.

Line 119. “First, the responses of axons in the upper block were, on average, more transient than those in the lower block (Fig. 3a-c, $p < 0.001$ by one-sided Wilcoxon rank sum, upper block: $n = 27$, lower block: $n = 20$, $r = 0.61$, $z = 4.19$).” Which size spots are being compared? What is the metric of transience?

We calculate transience based on the responses to square-wave-modulated 50, 100, and 200 μm spots. We include the formula for the calculation of transience in the Methods section of our revised manuscript (see also, below).

Response transience (T): $T = \frac{r_e - r_l}{r_e}$, where r_e represents the average responses to 50, 100, and 200 μm diameter spots early (0.23 – 0.67 s) after spot onset, and r_l refers to the average responses to the same stimuli later (1.21 – 1.65 s after spot onset).

Reviewer #4 (Remarks to the Author):

This manuscript reports on the response properties of ON-type bipolar cells in the mouse retina. First, the authors use sparse ON BC labeling and demonstrate that all ON BC types can be distinguished in their experiments based on anatomical features. They then use dense ON BC GCaMP6f expression to measure calcium responses in axonal arbors and compare response properties between apparent types at the population level. These analyses are well designed and type-specificity of the findings is credible.

Based on the measured responses, the authors functionally organize the complement of ON BC types into two blocks, distal and proximal, anatomically separated by the ON starburst amacrine cell dendritic layer. They argue that selective visual processing is distributed across these two blocks (distal: spatially uniform, temporally diverse; proximal: spatially diverse, temporally uniform), and this is reasonably supported by the data. To test generality of this claim and avoid bias from the use of artificial visual stimuli, the authors generated 'mouse-cam'-style videos and used those to stimulate the retina while recording ON bipolar cell population responses. Responses are analyzed with appropriate means including analysis of response correlations with feature representation-type transformations of the videos. The results of this analysis support the main claims. Finally, comparison of calcium responses within individual axonal branches of single cells showed uniform signaling across the axonal arbor. The noted absence of evidence for subcellular, localized computation is relevant following recent claims to the contrary in directional motion encoding.

A temporal division of signaling with transient responses near the central IPL and sustained responses proximal was already known but spatial tuning and cell-type specific detail was lacking. The comprehensive mapping done here resolves all of that, and also shows one exception to the temporal rule (the BC5t type is sustained within the presumed transient zone), and shows how spatial response properties co-organize with it. The study gives useful and apparent complete information about ON BC organization across IPL layers, and this is in my opinion is the main contribution of the work.

We thank the reviewer for carefully reading our manuscript, the clear summary, and the positive remarks. We address specific comments point-by-point below. We are grateful to the reviewer for helping us improve our manuscript.

Comments.

1. During stimulation with naturalistic movies, receptive field position within the imaged area mattered: because the movie was not spatially uniform, different parts of the image contain different information/features, and this would create differences in the responses of simultaneously recorded BCs at different spatial positions within the imaged field. How large was the imaged area? How did you account for the fact that not all BCs received the same visual stimulation? This must be addressed, for example in Results, around Line 195.

We thank the reviewer for bringing up this important point. For most of our recordings, we mapped ON bipolar cell receptive fields using binary checkerboard white noise stimuli. We found that 95% of the recorded cells had receptive field centers within 18.3 μm from the display center (distance from center: $0 \pm 9.9 \mu\text{m}$, mean \pm SD), their assumed position in the analyses mentioned by the reviewer. For our revisions, we calculated the spatial correlation of the information/features extracted from naturalistic movies for receptive fields pairs with increasing center-center distances. For all information/features, correlation coefficients (R^2) for center-center distances at the 95% boundary (i.e., 18.3 μm) were >0.9 . We conclude that deviations in receptive field center positions did not significantly affect our analysis. We have included results from the analysis

described above in a new Supplementary Figure (Supplementary Fig. 13) in our revised manuscript.

2. Line 245-246 is cryptic and should be elaborated: what does it mean that ‘motion provides an independent feature [...] that is distributed across ON BCs in both blocks’?

We thank the reviewer for bringing this to our attention. Coherent motion does not correlate significantly with temporal or spatial contrast in naturalistic movies, indicating that they are statistically independent features. To clarify this, we have rewritten the respective part of the Results section in our revised manuscript as follows: *“There was little correlation between coherent motion and temporal ($R^2 = 0.00014$) or spatial contrast ($R^2 = 0.023$), indicating that they are statistically independent features of naturalistic movies multiplexed in the stimulus encoding of the ON bipolar cell population.”*

3. Line 294-295, the claim that ‘for naturalistic movies, ON bipolar cells vary systematically in their responses to coherent motion’ needs to be better substantiated in Results, with references to the data/figures.

We show the cell-type-specific differences in the coherent motion sensitivity of ON bipolar cells in Supplementary Fig. 14 (panel c). In response to the reviewer’s comment, we have added the statistics as well as reference to Supplementary Fig. 14 to the respective section of the Results (Line 237-239) in our revised manuscript as follows: *“Consistent with this observation, coherent motion sensitivity varied across bipolar cell types in the upper and lower block (Supplementary Fig. 14, Kruskal Wallis one-way ANOVA, $p < 10^{-7}$, $\eta^2 = 0.24$).”*

Minor.

4. For ‘split into ON and OFF channels’ (line 35-36) consider including reference to Ratliff, ..., Balasubramanian, 2010, which demonstrates the benefit of the specific (uneven) ON-OFF split within the mammalian retina with regards to natural scene encoding.

Thanks, we have included this citation.

5. Fig. 1f legend text typo: ‘...profiles OF all ON bipolar... ‘.

We have corrected this typo.

6. Fig. 2B: colors of the traces representing different BC types are difficult to discern. Consider broader color scale and/or adding symbols.

We experimented with several color-scales for our revisions but ultimately kept our previous one, because we felt it was the clearest and best for color-blind readers. We hope the reviewer is okay with this choice.

7. Fig. 3a legend text, typo: ‘... to stimuli spots... ‘ correct.

We have corrected this mistake. Thanks for spotting it.

REVIEWERS' COMMENTS

Reviewer #1 (Remarks to the Author):

The authors have adequately addressed my previous concerns.

Reviewer #2 (Remarks to the Author):

The authors addressed all my previous concerns. It is an excellent paper.

Reviewer #3 (Remarks to the Author):

My suggestions/criticisms have been addressed adequately

Reviewer #4 (Remarks to the Author):

The authors have satisfactorily addressed my comments on the initial submission. The color scheme of figs 2 -5 is acceptable as is.

Reviewer #1:

The authors have adequately addressed my previous concerns.

We thank the reviewer for their positive assessment of our revised manuscript.

Reviewer #2:

The authors addressed all my previous concerns. It is an excellent paper.

We thank the reviewer for their positive assessment of our revised manuscript.

Reviewer #3:

My suggestions/criticisms have been addressed adequately.

We thank the reviewer for their positive assessment of our revised manuscript.

Reviewer #4:

The authors have satisfactorily addressed my comments on the initial submission. The color scheme of figs 2 -5 is acceptable as is.

We thank the reviewer for their positive assessment of our revised manuscript.